# Structural motifs for subtype-specific pH-sensitive gating of vertebrate otopetrin proton channels

**Bochuan Teng[1,2†], Joshua P Kaplan[1,2†], Ziyu Liang[1,2], Zachary Krieger[1], Yu-Hsiang Tu[3], Batuujin Burendei[4], Andrew B Ward[4], Emily R Liman[1,2]***

[1]Section of Neurobiology, Department of Biological Sciences, University of Southern California, Los Angeles, United States; [2]Program in Neuroscience, University of Southern California, Los Angeles, United States; [3]Diabetes, Endocrinology, and Obesity Branch, National Institute of Diabetes and Digestive and Kidney Diseases, National Institutes of Health, Bethesda, United States; [4]Department of Integrative Structural and Computational Biology, The Scripps Research Institute, La Jolla, United States

**\*For correspondence:**
liman@usc.edu

†These authors contributed equally to this work

**Competing interest:** The authors declare that no competing interests exist.

**Abstract** Otopetrin (OTOP) channels are proton-selective ion channels conserved among vertebrates and invertebrates, with no structural similarity to other ion channels. There are three vertebrate OTOP channels (OTOP1, OTOP2, and OTOP3), of which one (OTOP1) functions as a sour taste receptor. Whether extracellular protons gate OTOP channels, in addition to permeating them, was not known. Here, we compare the functional properties of the three murine OTOP channels using patch-clamp recording and cytosolic pH microfluorimetry. We find that OTOP1 and OTOP3 are both steeply activated by extracellular protons, with thresholds of $pH_o$ <6.0 and 5.5, respectively, and kinetics that are pH-dependent. In contrast, OTOP2 channels are broadly active over a large pH range (pH 5 pH 10) and carry outward currents in response to extracellular alkalinization (>pH 9.0). Strikingly, we could change the pH-sensitive gating of OTOP2 and OTOP3 channels by swapping extracellular linkers that connect transmembrane domains. Swaps of extracellular linkers in the N domain, comprising transmembrane domains 1–6, tended to change the relative conductance at alkaline pH of chimeric channels, while swaps within the C domain, containing transmembrane domains 7–12, tended to change the rates of OTOP3 current activation. We conclude that members of the OTOP channel family are proton-gated (acid-sensitive) proton channels and that the gating apparatus is distributed across multiple extracellular regions within both the N and C domains of the channels. In addition to the taste system, OTOP channels are expressed in the vertebrate vestibular and digestive systems. The distinct gating properties we describe may allow them to subserve varying cell-type specific functions in these and other biological systems.

## Editor's evaluation

The manuscript shows that OTOP proton channels are proton-gated with distinct pH sensitivities, and identifies regions on the proteins that alter pH-dependent gating. The main claims are well supported by the data. These findings are likely to be of interest to researchers studying acid/base physiology, sensory physiology, and ion channel biophysics.

## Introduction

The activity of ion channels is often tightly controlled through a change in conformation, known as gating, that opens and closes the ion permeation pathway (*Hille, 2001*). Gating can be in response to voltage (voltage-dependent), chemical stimuli (ligand-gated), or membrane deformation

(mechanically-gated). Understanding the gating of an ion channel is critical to understanding its physiological function. Recently, a new family of ion channels that are selective for protons and with little or no structural similarity to other ion channels was identified, collectively named the Otopetrins or OTOPs (*Tu et al., 2018*). These ion channels mediate proton influx in response to acid stimuli, but whether protons also gate them was unknown. Notably, while the structures of vertebrate OTOP1 and OTOP3 channels were recently solved (*Chen et al., 2019*; *Saotome et al., 2019*), it is not known if these structures are in open or closed states due to the lack of information regarding the gating of the channels.

OTOP1 currents were first characterized in taste receptor cells, where pH sensitivity, proton selectivity, and inhibition by $Zn^{2+}$ were described (*Chang et al., 2010*; *Bushman et al., 2015*). The founding member of the OTOP family, mOTOP1, was identified as the product of a gene mutated in a murine vestibular disorder (*Hurle et al., 2003*; *Hughes et al., 2004*) and was subsequently shown to form a proton channel that functions as a receptor for sour taste in vertebrates (*Tu et al., 2018*; *Teng et al., 2019*; *Zhang et al., 2019*). Most vertebrate genomes encode two related proteins, OTOP2 and OTOP3, that also form proton channels (*Tu et al., 2018*) and are expressed in a diverse array of tissues, including in the digestive tract, where mutations in the corresponding genes have been linked to disease (*Tu et al., 2018*; *Parikh et al., 2019*; *Qu et al., 2019*; *Yang et al., 2019a*). Functional OTOP channels are conserved across species, including in invertebrates, where they play roles in acid sensing and biomineralization (*Hurle et al., 2011*; *Tu et al., 2018*; *Chang et al., 2021*; *Ganguly et al., 2021*; *Mi et al., 2021*).

Ion channels selective for protons are rare in nature, comprising just a small subset of the hundreds of types of ion channels that have been described over the last 80 years (*Hille, 2001*). The two best characterized proton-selective ion channels are M2, a viral protein involved in the acidification of the influenza virus interior, and Hv1, a voltage-gated ion channel that extrudes protons during the phagocyte respiratory burst to maintain pH (*Pinto et al., 1992*; *Ramsey et al., 2006*; *Sasaki et al., 2006*; *Morgan et al., 2009*). OTOP1 channels, unlike HV1, are not gated by voltage (*Bushman et al., 2015*; *Tu et al., 2018*) but might instead be gated by protons, like Hv1 and M2 (*Cherny et al., 1995*; *Liang et al., 2016*). However, establishing that the permeant ion gates an ion channel is not trivial. For example, an increase in current magnitudes as pH is lowered could be attributed to the opening of channels or an increase in the driving force for proton entry. Here we describe the biophysical response properties of the three murine OTOP channels to varying extracellular pH. By focusing on parameters that are independent of the driving force, we show that extracellular protons gate the three channels in a subtype-specific manner and that the extracellular linker between transmembrane domains eleven and twelve plays a role in gating.

## Results

### Differential current response profiles of three murine OTOP channels to acidic and basic stimuli

We previously reported that all three murine OTOP channels conduct inward proton currents in response to acid stimuli, but show differences in their current as a function of pH (I-pH) relations (*Tu et al., 2018*). We reasoned that these differences likely reflect differences in gating. To test this hypothesis, we performed a careful comparative analysis of the response properties of murine OTOP1, OTOP2, and OTOP3.

We used patch-clamp recording from HEK-293 cells transfected with cDNAs encoding each of the three channels for these experiments. If not otherwise stated, the intracellular solution was pH 7.4 (Cs-Aspartate-based), and the holding potential was –80 mV. All three channels carried inward currents in response to lowering $pH_o$ in a $Na^+$-free solution, as previously reported (*Tu et al., 2018*). The magnitude of the OTOP1 currents increased as the extracellular pH ($pH_o$) was lowered over a range of pH 6 to pH 4.5, while the magnitude of OTOP2 currents changed very little over the same pH range (*Figure 1A*). OTOP3 currents increased more steeply over the same range, with little or no inward currents in response to pH 6 (*Figure 1A and C*). For all three channels, currents decayed in response to prolonged acid exposure, and the rate of current decay was faster as the $pH_o$ was lowered (*Figure 1A*), likely due in part to intracellular accumulation of protons (*DeCoursey and Cherny, 1996*; *Bushman et al., 2015*; *De-la-Rosa et al., 2016*; *Tu et al., 2018*) (discussed below).

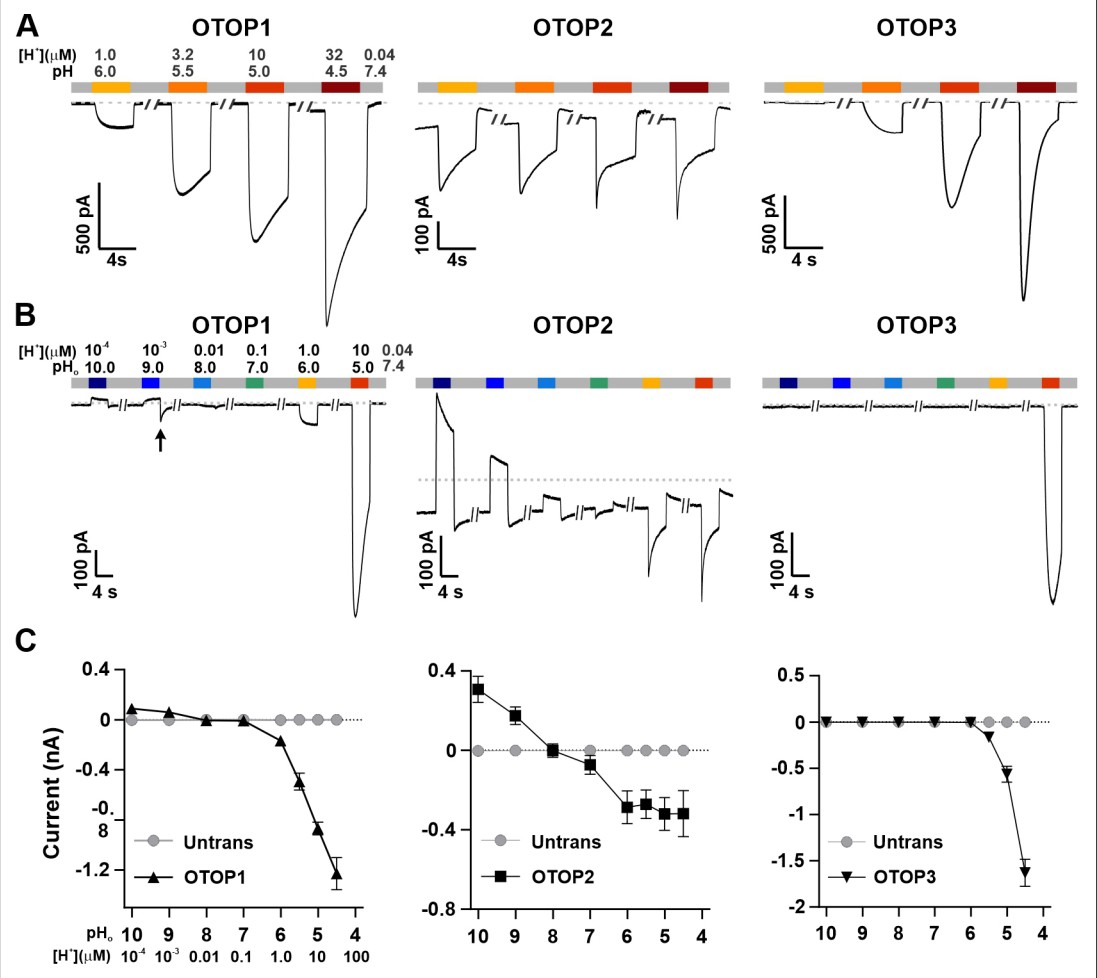

**Figure 1.** Three vertebrate OTOP channels vary in their I-pH response profile across a broad pH range. (**A**) Proton currents elicited in response to acidic stimuli (pH 6–4.5) in the absence of extracellular $Na^+$ measured from HEK-293 cells expressing each of the three OTOP channels as labeled ($V_m$ = -80 mV). (**B**) Proton currents in response to solutions that varied in pH (pH 10–5) were measured from HEK-293 cells expressing each of the three OTOP channels as labeled ($V_m$ = -80 mV). (**C**) Average data showing the peak current magnitude in response to stimuli of varying $pH_o$, measured from experiments as in (**A**) and (**B**). OTOP3 currents are only observed in response to solutions of pH <5.5, while OTOP1 currents are evoked by solutions of <pH 6 and alkaline stimuli, and OTOP2 currents are evoked in response to all stimuli. Data represent mean ± s.e.m. of biological replicates where for each data point n=5–10 for OTOP1, n=6–10 for OTOP2, and n=4–10 for OTOP3.

The online version of this article includes the following source data for figure 1:

**Source data 1.** Source data for *Figure 1C*.

To further investigate the I-pH response profile of the three channels, we extended the pH range of the test solutions, now including neutral and alkaline solutions (*Figure 1B and C*). In OTOP1-expressing cells, we observed a small outward current in response to the alkaline solutions (pH 9–10) that was not observed in untransfected cells, which we, therefore, attribute to currents through OTOP1 channels. In response to the pH 9, but not pH 10 solution, we observed a 'tail current' upon return to neutral pH (for pH 9, $I_{tail}$ = –123+/-20 pA, n=5; for pH 10, $I_{tail}$ = –36+/-11 pA); this may reflect cytosolic $H^+$ depletion during the pH 9 stimulus, creating a driving force for proton entry through open OTOP1 channels that subsequently close when the pH was restored to 7.4. In response to pH 8 and 7 solutions, no measurable change in the baseline currents was observed.

A very different pH-response profile was observed in OTOP2-expressing cells (*Figure 1B and C*). Quite surprisingly, large outward currents were evoked in response to alkaline solutions of pH 9 and pH 10, which were similar in magnitude to the inward currents evoked in response to the acidic solutions. We also observed changes in the holding current in response to solutions at near-neutral pH (pH 8 or 7), suggesting that the channels are open at the resting $pH_o$. Overall, we observed changes

in OTOP2 currents in response to changes in extracellular pH over the entire pH range tested. The magnitude and direction of the responses were generally proportional to the driving force on the proton. This relationship broke down over the more acidic pH range, where a change of 10-fold in ion concentration (e.g. pH 6 versus pH 5; *Figure 1*) did not lead to a substantial increase in current magnitude. Although we do not yet understand this phenomenon, we suspect that it may be due in part to the rapid decay kinetics of the currents in response to acid stimuli and the consequent attenuation of the peak currents as well as possible inhibition of the channels by intracellular acidification.

OTOP3 currents were active over a narrower pH range than OTOP1 and OTOP2 currents (*Figure 1B and C*). They were activated steeply below pH 6, and no outward currents were observed in response to any of the alkaline stimuli.

These data suggest a sharp threshold for activating OTOP1 and OTOP3 channels of ~pH 6 (OTOP1) and pH 5.5. (OTOP3). In contrast, OTOP2 channels are active over the entire pH range. They also show that both OTOP1 and OTOP2 channels can conduct outward currents, which may be physiologically relevant under some circumstances.

## OTOP2 is selective for protons and open at neutral pH

The unusual response properties of OTOP2 raise the question of whether OTOP2 is selective for protons. OTOP1 is highly selective for $H^+$ over $Na^+$, by a factor of at least $10^5$ fold (*Tu et al., 2018*), and currents carried by OTOP3 follow the expectations for a proton-selective current, such as a shift

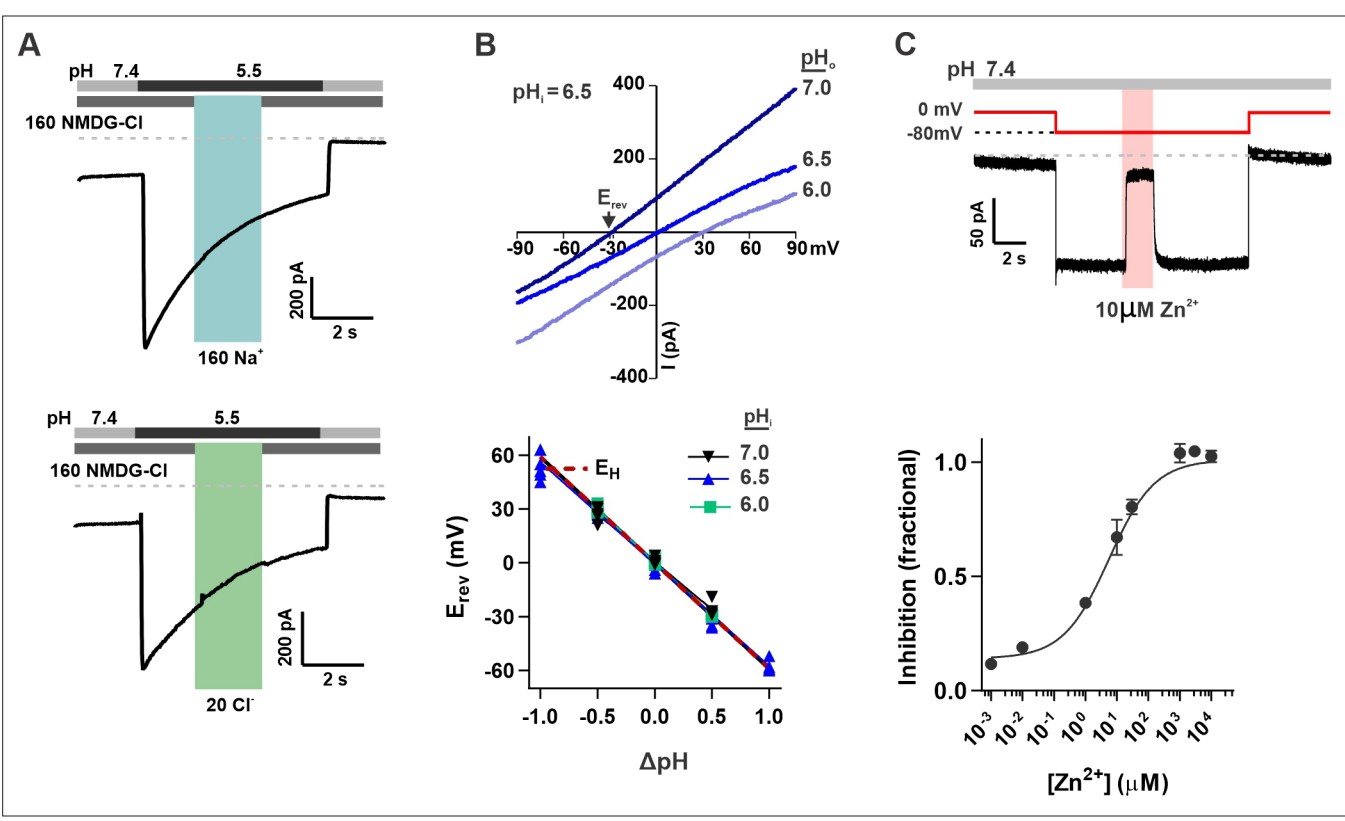

**Figure 2.** OTOP2 is a proton selective ion channel that is open at neutral $pH_o$. (**A**) Representative OTOP2 current elicited in response to a pH 5.5 solution with 160 mM $Na^+$ replacing $NMDG^+$ (top) or 140 mM Methane sulfonate⁻/ 20 mM $Cl^-$ replacing 160 mM $Cl^-$ (bottom) in the extracellular solutions at times indicated. $V_m$ was held at –80 mV. (**B**) The I-V relationship (top) of the $Zn^{2+}$ sensitive component of OTOP2 current in response to different $pH_o$ stimuli was obtained from ramp depolarizations in the presence and absence of $Zn^{2+}$. $pH_i$ was adjusted to 6.5. Bottom: $E_{rev}$ measured as a function of $\Delta pH$ ($pH_o – pH_i$). $pH_i$ was adjusted to 6.0, 6.5, or 7.0 as indicated. The red dotted line is the predicted equilibrium potential for $H^+$, $E_H$. (**C**) Extracellular $Zn^{2+}$ inhibits resting OTOP2 currents in a dose-dependent manner. Trace (top) shows inhibition of resting current in OTOP2-expressing HEK-293 cells by 10 μM $Zn^{2+}$. $V_m$ as indicated. Fractional inhibition was fit with a Hill slope = 0.6 and $IC_{50}$=5.6 μM. Data represent mean ± s.e.m of biological replicates where n=3–4 for each data point.

The online version of this article includes the following source data for figure 2:

**Source data 1.** Source data for *Figure 2B and C*.

in the reversal potential that follows the equilibrium potential for the H⁺ ion (*Tu et al., 2018*; *Chen et al., 2019*). OTOP2 is known to permeate protons (*Tu et al., 2018*), but selectivity for protons was not previously measured. To address this question, we first performed ion substitution experiments. We observed no change in the magnitude of the currents when the concentrations of either $Na^+$ or $Cl^-$ in the extracellular solution were changed (*Figure 2A*), indicating that OTOP2 is not permeable to either ion. We also measured the reversal potential of OTOP2 currents as a function of ΔpH ($pH_o – pH_i$) under conditions designed to minimize H⁺ ion accumulation (see methods) (*Cherny and DeCoursey, 1999*; *Ramsey et al., 2006*; *Tu et al., 2018*). These experiments showed that $E_{rev}$ changed ~59 mV/pH as expected for an H⁺-selective ion channel (*Figure 2B*). Thus, we conclude that OTOP2 is selective for protons.

The unusual response properties of OTOP2 suggested that it might be open when $pH_o$ = 7.4. Consistent with this possibility, we routinely observed a large resting current at –80 mV in OTOP2-expressing cells. We reasoned that if this resting current is due to open OTOP2 channels, it should be inhibited by $Zn^{2+}$, a blocker of OTOP1 and other proton channels (*Tu et al., 2018*). Indeed, OTOP2 currents at neutral pH were inhibited by $Zn^{2+}$ with an $IC_{50}$ of 5.6 µM (*Figure 2C*). Note that because the inhibition of OTOP1 by $Zn^{2+}$ is pH-dependent (*Bushman et al., 2015*; *Teng et al., 2019*), the sensitivity of OTOP2 currents at neutral pH to micromolar concentrations $Zn^{2+}$ as compared with the requirement of millimolar concentrations of $Zn^{2+}$ to inhibit OTOP1 currents evoked in response to acid stimuli is as to be expected.

Thus, we concluded that OTOP2 is a proton-selective ion channel open at neutral pH.

## Different pH-response profiles of three murine OTOP channels measured with pH imaging

Inward and outward currents carried by protons through OTOP channels are expected to cause a change in intracellular pH. Thus, to confirm the results described above, we assessed the activity of each OTOP channel by monitoring intracellular pH. For these experiments we co-transfected HEK-293 cells with a pH-sensitive fluorescent protein – pHluorin, a variant of GFP whose fluorescence emission changes as a function of the intracellular pH (*Miesenböck et al., 1998*). As a control to confirm the expression and activity of pHluorin, all cells were exposed at the end of the experiment to acetic

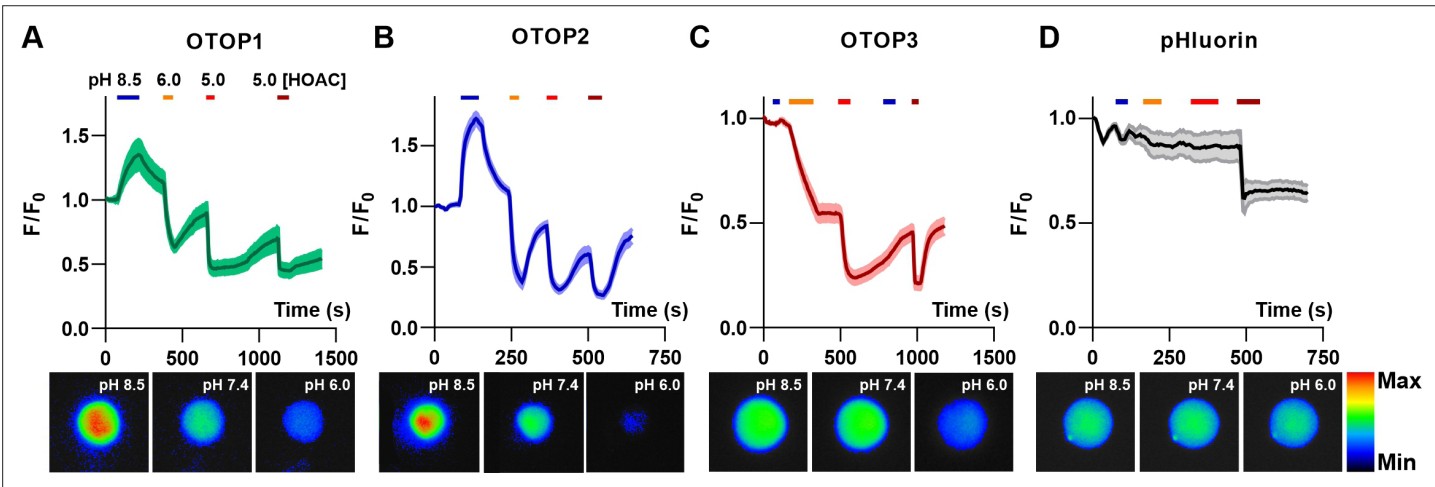

**Figure 3.** OTOP channels mediate the influx and efflux of protons as measured with intracellular pH imaging. (**A–D**) Changes in intracellular fluorescence emission upon exposure to changing extracellular pH, as indicated, were measured from HEK-293 cells co-expressing OTOP channels and the pH-sensitive indicator pHluorin (**A–C**) or pHluorin alone (**D**). Data are shown as the mean ± SEM for n=7, 7, 11, and 3 cells for (**A**), (**B**), (**C**), and (**D**), respectively. Acetic acid, which is permeable through cell membranes and acidifies cell cytosol directly, served as a positive control. Only OTOP1 and OTOP2 conducted protons out of the cell cytosol in response to alkalinization, while all three channels conducted protons into the cell cytosol in response to acidification, albeit at different rates. The lower panel shows images of a single cell in the field of view used for these experiments taken at the pH indicated (pseudo color is arbitrary units).

The online version of this article includes the following source data for figure 3:

**Source data 1.** Source data for *Figure 3A–D*.

acid, which crosses cell membranes and causes intracellular acidification (*Figure 3A-D*, dark red bar) (*Wang et al., 2011*).

Cells expressing each of the three OTOP channels responded to the acidic solutions with large changes in intracellular pH, a response not observed in control cells (*Figure 3A* – D). Strikingly, we also observed changes in intracellular pH in response to an alkaline solution (pH$_o$ = 8.5) in both OTOP1- and OTOP2-expressing cells but not in OTOP3-expressing cells. This data is consistent with the patch-clamp data showing that OTOP1 and OTOP2, but not OTOP3, conduct outward proton currents in response to alkaline extracellular stimuli.

We also noted differences in the time course of the response to acidic solutions (pH 6 or pH 5) in cells expressing the three OTOP channels. Notably, the response of OTOP3-expressing cells appeared slower than that of OTOP1 or OTOP2-expressing cells (compare *Figure 3C* with 3 A and 3B). This is likely a consequence of the smaller currents that are evoked by these stimuli in OTOP3 cells and the slower kinetics of channel activation (see below).

Together these data support the conclusion that OTOP1 and OTOP2 can conduct outward proton currents in response to alkaline extracellular solution solutions, while all three channels carry inward proton currents in response to acidic stimuli.

## Kinetics of OTOP channels provide direct evidence of pH-dependent gating

In patch-clamp recordings such as those shown in *Figure 1*, we noted differences in the kinetics of the currents elicited in response to acidic solutions among the three channels. For example, OTOP2 currents showed faster activation than OTOP1 currents, while OTOP3 currents were considerably slower. These differences likely reflect differences in activation of the channels by protons. In particular, for a channel that is open at neutral pH (OTOP2), we expect the currents to change in response to an increase or decrease in the concentration of the permeant ion (H$^+$) with a time course that reflects the speed of the solution exchange. In contrast, for a channel that is closed at neutral pH (OTOP1 and OTOP3), currents may increase more slowly, with kinetics determined by the rates of agonist binding and opening of the channels.

The rate of activation of each channel in response to acid stimuli was measured by fitting the time course of the current upon solution exchange with a single exponential (*Figure 4A*), excluding the first few milliseconds where responses deviated from an exponential time course due to a lag in the solution exchange. We also measured the rate of decay of the currents after return to neutral pH in a similar manner.

In response to the pH 5.5 stimulus, the activation kinetics of the three OTOP channels varied over nearly two orders of magnitude (*Figure 4A and C*). The time constant for activation of OTOP2 was 12.0±1.6ms (n=10) which was slightly faster than the rate of solution exchange as measured using cells expressing an inward rectifier K$^+$ channel and solutions that varied in potassium concentration (*Figure 4B and C*; $\tau$=26.1 ± 2.9ms). Time constants for activation of OTOP1 and OTOP3 were 142.7±13.5ms (n=6) and 1098.0±83.0ms (n=6), respectively, considerably slower than the solution exchange suggesting that both are gated by extracellular protons. In contrast, the time constant for deactivation of OTOP1, OTOP2, and OTOP3 currents were similar to the rate of the solution exchange ($\tau$=18.6 ± 4.4ms, n=6, 19.3±2.8ms, n=9, and 20.9±3.0ms, n=13, respectively; *Figure 4A and C*). The off-rates of the OTOP currents likely reflect the rate of removal of the permeant ion and not the closing rates of the channels (which could be slower). This also confirms that the differences in on-rates were not artifacts of varying solution exchange times between groups of cells.

We also measured the kinetics of the currents carried by all three channels in response to solutions that varied in pH$_o$. The activation rate of OTOP1 and OTOP3 currents increased as the pH$_o$ was lowered (*Figure 4D* and *Figure 4—figure supplement 1*). This is as expected if protons gate the channels. In contrast, the activation rate of the OTOP2 currents was insensitive to the extracellular pH over this range, consistent with an interpretation that the channels are open at neutral pH, and lowering the pH does not further increase channel open probability (*Figure 4D* and *Figure 4—figure supplement 1*). The decay rate of the currents upon return to neutral pH did not vary as a function of the pH of the stimulus for any of the three channels (*Figure 4D*).

Together, the slow, pH-dependent kinetics of currents carried by OTOP1 and OTOP3 channels provides strong evidence that they are gated by extracellular protons.

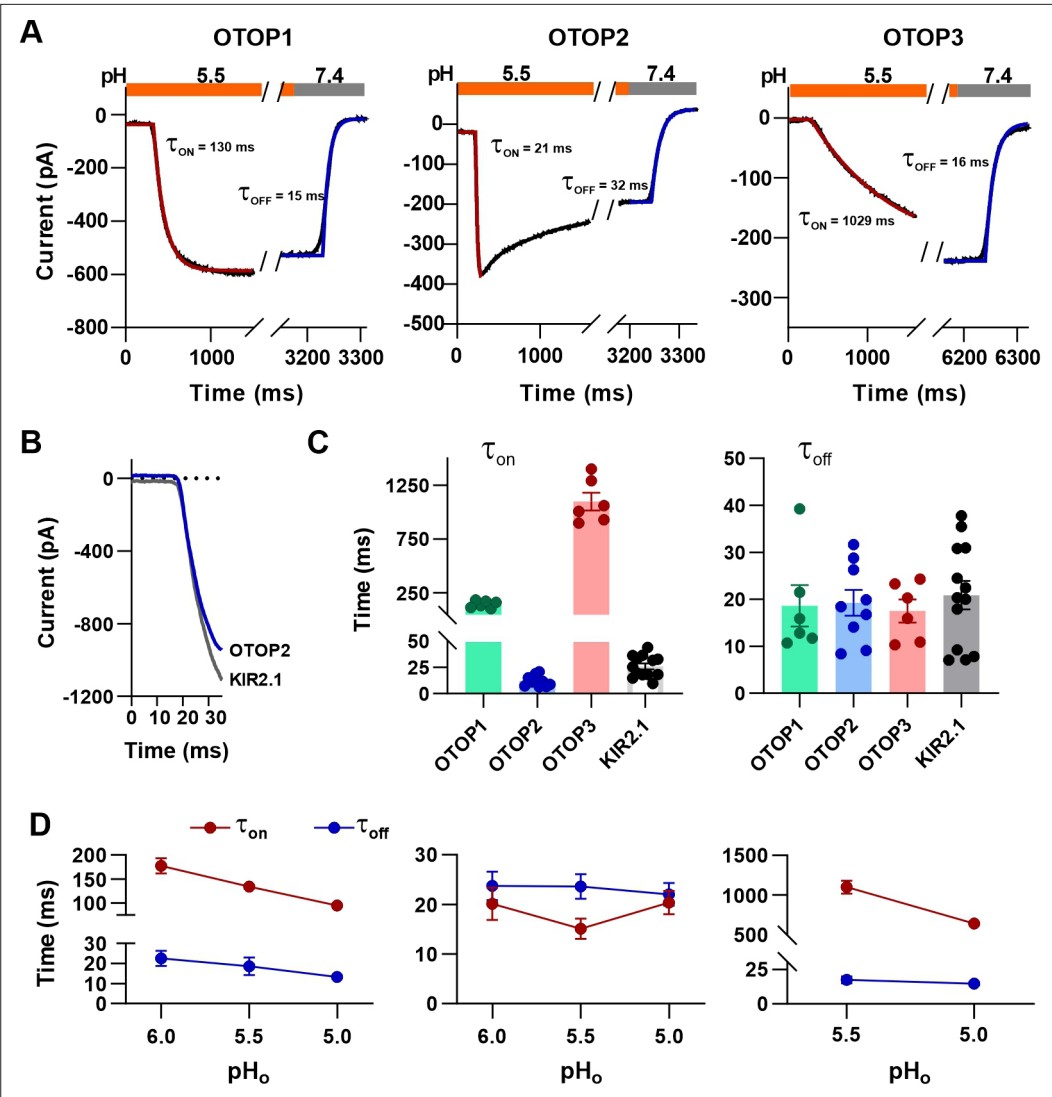

**Figure 4.** Activation kinetics vary dramatically between OTOP channels. (**A**) Representative current traces (black) from cells expressing each of the three OTOP channels in response to the application and the removal of pH 5.5 solutions. The activation and the decay kinetics of the currents were fitted with a single exponential (red and blue curves, respectively). (**B**) Solution exchange kinetics measured with an open $K^+$ channel KIR2.1 are similar to the kinetics of OTOP2 currents. (**C**) Summary data for $\tau_{on}$ (left panel) and $t_{off}$ (right panel) of the three OTOP channels (OTOP1: green, OTOP2: blue, OTOP3: red) and KIR2.1 (black) in response to the application and removal of pH 5.5 extracellular solution. n=6, 5–6, 6 for OTOP1, OTOP2 and OTOP3 respectively. (**D**) Time constants for activation ($\tau_{on}$, red) and deactivation ($\tau_{off}$, blue) of OTOP1, OTOP2, and OTOP3 currents in response to acidic stimuli (pH 6.0, 5.5, and 5.0) measured from experiments as in A. Note that the data at pH 5.5. is also shown in Panel C. n=6,5–6, and 6 for OTOP1, OTOP2, and OTOP3, respectively.

The online version of this article includes the following source data and figure supplement(s) for figure 4:

**Source data 1.** Source data for *Figure 4C and D*.

**Figure supplement 1.** Fit to activation kinetics of OTOP currents.

## Effect of extracellular pH on slope conductance

To provide further evidence that extracellular protons gate OTOP channels, we measured the slope conductance as a function of $pH_o$ for all three channels. The slope conductance ($\Delta I/\Delta V$) was measured from responses to ramp depolarizations at or near the peak of the response to the stimulus ($pH_o$ = 10–5) (*Figure 5A and B*) using a range of $V_m$ from –80 mV to 0 mV, where the currents were generally linear and not contaminated by endogenous currents. Slope conductance (G) is related to the

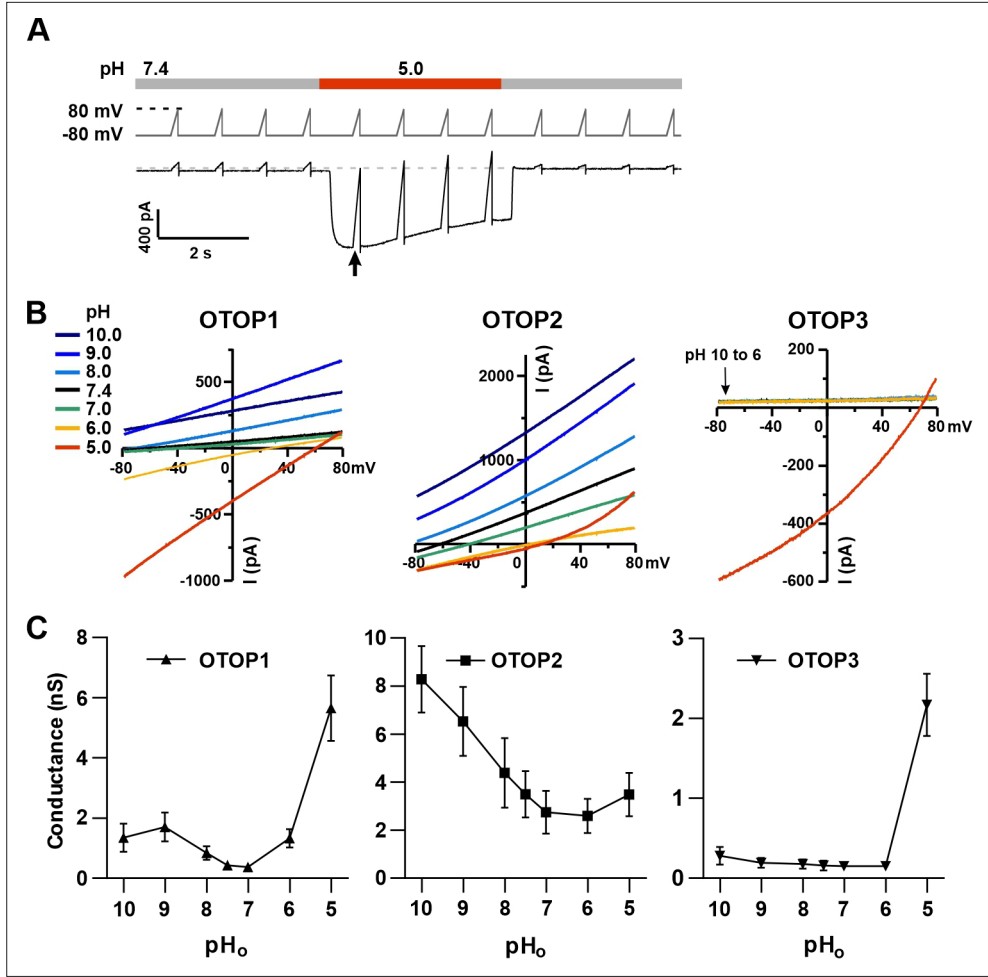

**Figure 5.** Changes in the slope conductance of OTOP channels as a function of extracellular pH. (**A**) Voltage and solution exchange protocol used to measure the slope conductance in response to changing extracellular pH. $V_m$ was held at –80 mV and ramped to +80 mV (1 V/s at 1 Hz). The first ramp after the currents peaked was used for later measurements. (**B**) Representative I-V relationship from HEK-293 cells expressing each of the three OTOP channels in response to alkaline or acidic stimuli (pH 10–5) from experiments described in (**A**). The conductance was measured from the slope of the I-V curve between –80 mV and 0 mV to avoid contamination from outwardly rectifying Cl⁻ currents. (**C**) Average slope conductance measured from cells expressing each of the three OTOP channels in response to different $pH_o$ stimuli from data as in (**B**). Data represent mean ± s.e.m. of biological replicates where for each data point n=5 for OTOP1, n=6–7 for OTOP2, and n=5–7 for OTOP3.

The online version of this article includes the following source data and figure supplement(s) for figure 5:

**Source data 1.** Source data for *Figure 5C*.

**Figure supplement 1.** Changes in reversal potentials of OTOP currents during prolonged exposure to acid stimuli.

**Figure supplement 1—source data 1.** Source data for *Figure 5—figure supplement 1B, C*.

---

channel's open probability ($P_o$) by G=N*g*$P_o$, where N is the number of channels and g is the single-channel conductance. Since the number of channels is constant, the slope conductance is a measure of $P_o$*g. Importantly, this provides a measure of open probability, even when the holding potential is close to the equilibrium potential for H⁺ where currents are small (e.g. for stimuli at 8.0).

In OTOP1-expressing cells, the slope conductance varied as a function of extracellular pH, dramatically increasing when the extracellular pH was lowered below $pH_o$ = 6.0 (**Figure 5B and C**). We also observed an increase in the slope conductance when the $pH_o$ was made more alkaline, consistent with data in **Figure 1** showing that an alkaline $pH_o$ of 9.0 can be activating. In OTOP2-expressing cells, the slope conductance was highest in alkaline pH ($pH_o$ = 10) and decreased as the $pH_o$ was lowered

(*Figure 5B and C*). The response of OTOP3-expressing cells was more similar to that of OTOP1, but with some clear differences (*Figure 5B and C*). The slope conductance remained very small when extracellular pH was between 10 and 6 and increased drastically when $pH_o$ was 5. These data support the conclusion that OTOP1 and OTOP3 are gated by extracellular protons, while OTOP2 is conductive over a broad pH range.

This series of experiments also provided an opportunity to test the extent to which intracellular acidification accounts for the decay of the macroscopic currents. In particular, we noted that the decay of the currents was often accompanied by a shift in $E_{rev}$ (*Figure 5—figure supplement 1A*, B). Although in these experiments the currents were not leak-subtracted and, consequently, $E_{rev}$ cannot be used to precisely measure ΔpH, shifts in $E_{rev}$ can, nonetheless, be considered as evidence of a relative change in intracellular pH. For OTOP1, the observation that $E_{rev}$ shifted by –42.5+/-10.4 mV (n=5) between the first and the fourth ramp depolarization (4 seconds apart) suggests that the cytosol acidified by ~0.7 pH units (*Figure 5—figure supplement 1B*). A more modest shift of –27.9+/-3.4 mV (n=3) was observed for OTOP2 currents, while for slowly activating OTOP3 currents, $E_{rev}$ shifted slightly but significantly in the positive direction (6.82+/-2.5 mV, n=7; p=0.03, two-tailed paired Student's T-test comparing t=0 s and t=4 s). To test whether acidification of the cytosol could account for the decay of OTOP1 and OTOP2 currents, we calculated the expected effect of the change in driving force ($V_m$-$E_{rev}$) on the current magnitude at 1 s timepoints as the currents decayed and compared this to the observed current magnitudes at these times. As seen in *Figure 5—figure supplement 1C*, the shift in driving force mostly accounts for the decay of OTOP1 currents. For OTOP2, the decays in current magnitudes is much greater than can be explained by a change in driving force, suggesting that intracellular acidification may directly inhibit OTOP2 channel activity or conductance. We could not perform a similar analysis of OTOP3 currents, which in these experiments did not decay significantly during the stimulus.

## Extracellular linkers are key determinants for pH-sensitive gating

The difference in $pH_o$ sensitivity of the three murine OTOP channels suggested that structural domains that vary among the channels contribute to the gating apparatus and that these domains could be identified using chimeric channels. Specifically, we generated chimeras between mOTOP2 and mOTOP3, which, as described above, are the most divergent functionally of the three murine OTOP channels. OTOP channels contain twelve transmembrane domains (S1-S12), with N and C termini located intracellularly. The transmembrane domains 1–6 and 7–12 respectively constitute the structurally homologous 'N' and 'C' domains. Reasoning that the residues involved in sensing the extracellular pH and gating the channels would be located on the extracellular surface of the channels, we swapped each of the six extracellular linkers that connect transmembrane helices (*Figure 6—figure supplement 1*). Each of the twelve chimeric channels was then tested over a range from pH 10 to pH 5. To simplify the analysis, we divided the chimeras into four categories: OTOP2 N domain (OTOP2 backbone with OTOP3 linkers), OTOP2 C domain, OTOP3 N domain, and OTOP3 C domain.

Of the three OTOP2 N domain chimeras, two were functional. Strikingly, the replacement of the OTOP2 S5-S6 linker (L5-6) with that from OTOP3 nearly eliminated the outward currents in response to alkaline stimuli and reduced the response to the mildly acidic stimulus (pH 6) but had little effect on the magnitude or kinetics of the inward currents elicited in response to the pH 5 stimulus (*Figure 6A–C*; OTOP2/OTOP3(L5-6)). Replacement of the OTOP2 S1-2 linker with that from OTOP3 reduced current magnitudes (possibly due to effects on trafficking) but did not significantly change relative responses to the stimuli of varying pH. In contrast to the OTOP2 N domain chimeras, the two functional C domain chimeras (L7-8 and L11-12) showed a similar pH dependence as WT OTOP channels, suggesting that the mutations did not specifically affect channel gating (*Figure 6A and D–E*). None of the chimeras showed a significant change in kinetics (*Figure 7A, B and E*).

Next, we examined OTOP3 channels containing extracellular linkers from OTOP2. Of these, the most striking was an OTOP3 N domain chimera with the L3-4 of OTOP2, which conducted outward currents in response to extracellular alkalinization (*Figure 6F–H*). Moreover, this chimera activated at a more mild pH than OTOP3 channels and with faster kinetics (*Figure 7C and F*). Thus, the simple swap of the 3–4 linker conferred a partial gating phenotype of the donor, OTOP2. The chimera containing the L5-6 from OTOP2 was more similar to OTOP3, while the L1-2 chimera appeared to be nonfunctional. Interestingly, the C domain chimeras also produced apparent changes in gating (*Figure 7D*

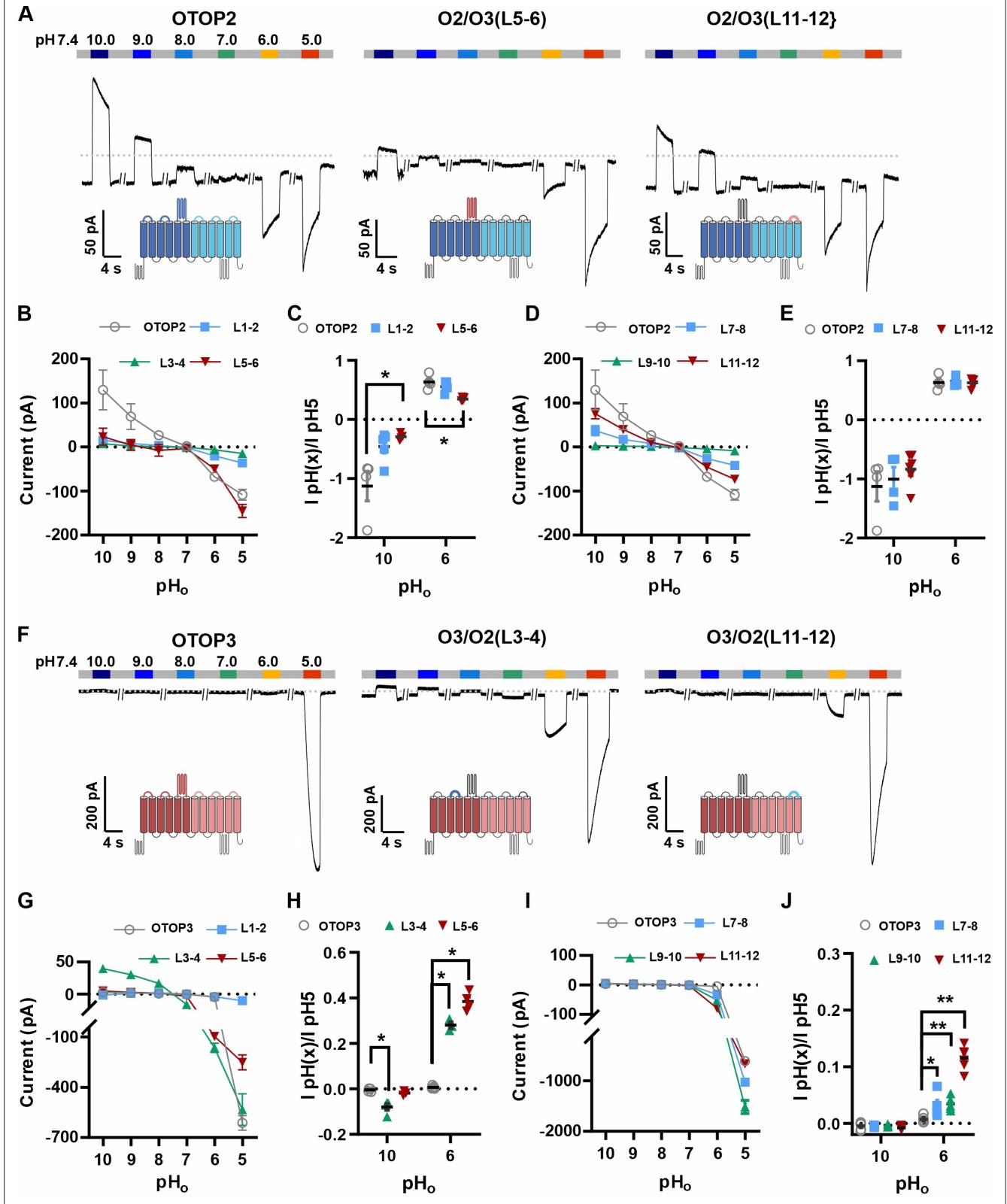

**Figure 6.** Chimeric channels with external linkers swapped reveal potential gating modules. (**A-F**) Representative current traces in response to varying pH from 10 to 5 measured from HEK-293 cells expressing OTOP2 channels and chimeras with an OTOP2 backbone (**A**) or OTOP3 channels and chimeras with an OTOP3 backbone (**F**) where numbers refer to linkers between corresponding transmembrane domains. Membrane potential was held at –80 mV. (**B, D, G, I**) Current magnitude (mean +/-s.e.m.) for OTOP2 N-domain chimeras (n=4–5) (**B**), OTOP2 C-domain chimeras (n=4–10) (**D**), OTOP3

*Figure 6 continued*

N-domain chimeras (n=3–5) (**G**), and OTOP3 C-domain chimeras (n=4–6) (**I**) from experiments such as in (**A**) and (**F**). (**C, E, H, J**) Same data for pH 10 and pH 6, normalized to the response to pH 5.0 to control for differences in expression. Significance was tested using the Mann-Whitney test. The p-values and n are given in *Figure 6—source data 2*.

The online version of this article includes the following source data and figure supplement(s) for figure 6:

**Source data 1.** Source data for *Figure 6B–E and G–J*.

**Source data 2.** Statistical tests comparing chimeric channels with wildtype channels with Mann-Whitney U test.

**Figure supplement 1.** Topology and sequence alignment of OTOP channels.

*and F*): all three chimeras activated more rapidly and at higher pH, adopting some features from the donor channel (OTOP2). But they retained the steep pH sensitivity of the OTOP3 channels and did not support proton efflux in response to alkaline stimuli like OTOP2.

The structure of OTOP2 channels has not yet been resolved experimentally, and the cryo-EM structures of OTOP1 and OTOP3 channels do not allow the visualization of the S5-6 linker (*Chen et al., 2019*; *Saotome et al., 2019*), possibly because the linker adopts multiple conformations. Thus,

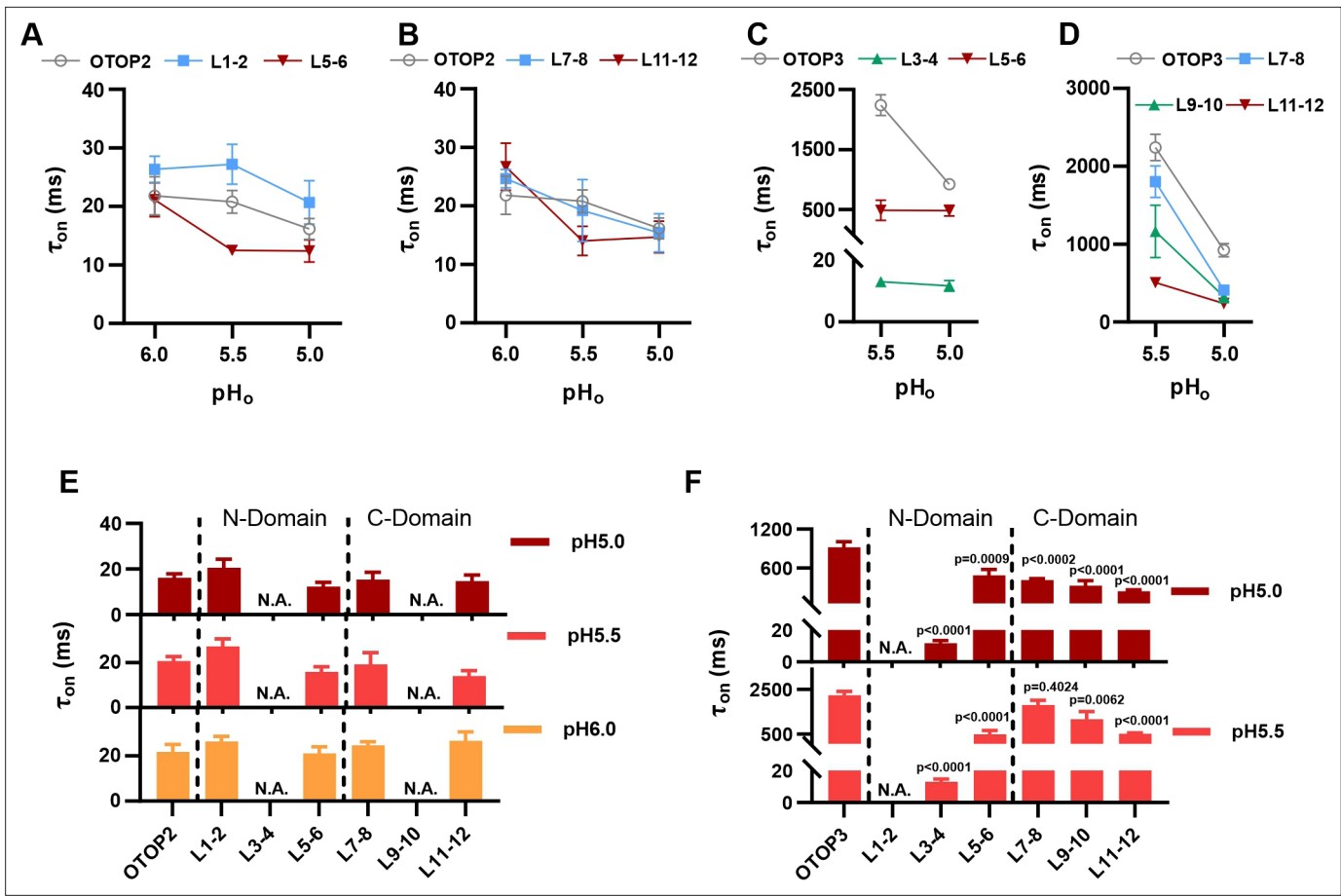

**Figure 7.** Increase in activation kinetics of mOTOP3 chimeric channels. (**A–D**) Time constants for activation of chimeric as compared with wildtype channels, measured from traces as in *Figure 6A and F*, using methods as in *Figure 4*. The $\tau_{on}$ of OTOP2 and its chimeras (**A, B**) was not pH dependent and followed the rate of the solution exchange. Data are mean ± s.e.m, (n=3–4). The $\tau_{on}$ of OTOP3 chimeras bearing OTOP2 linkers (**C, D**) was generally faster than that of the wildtype channels (n=3–4). Same nomenclature as in *Figure 6*. (**E, F**) Same data as in A-D, plotted to allow comparison across all chimeras at the same pH (as indicated; note that because wildtype OTOP3 is not activated by pH 6, data using a pH 6 stimulus is not included in the analysis). Statistical results report the comparison between chimeric channels and the wildtype channel using ANOVA and Dunnett's multiple comparisons tests.

The online version of this article includes the following source data for figure 7:

**Source data 1.** Source data for *Figure 7A–D*.

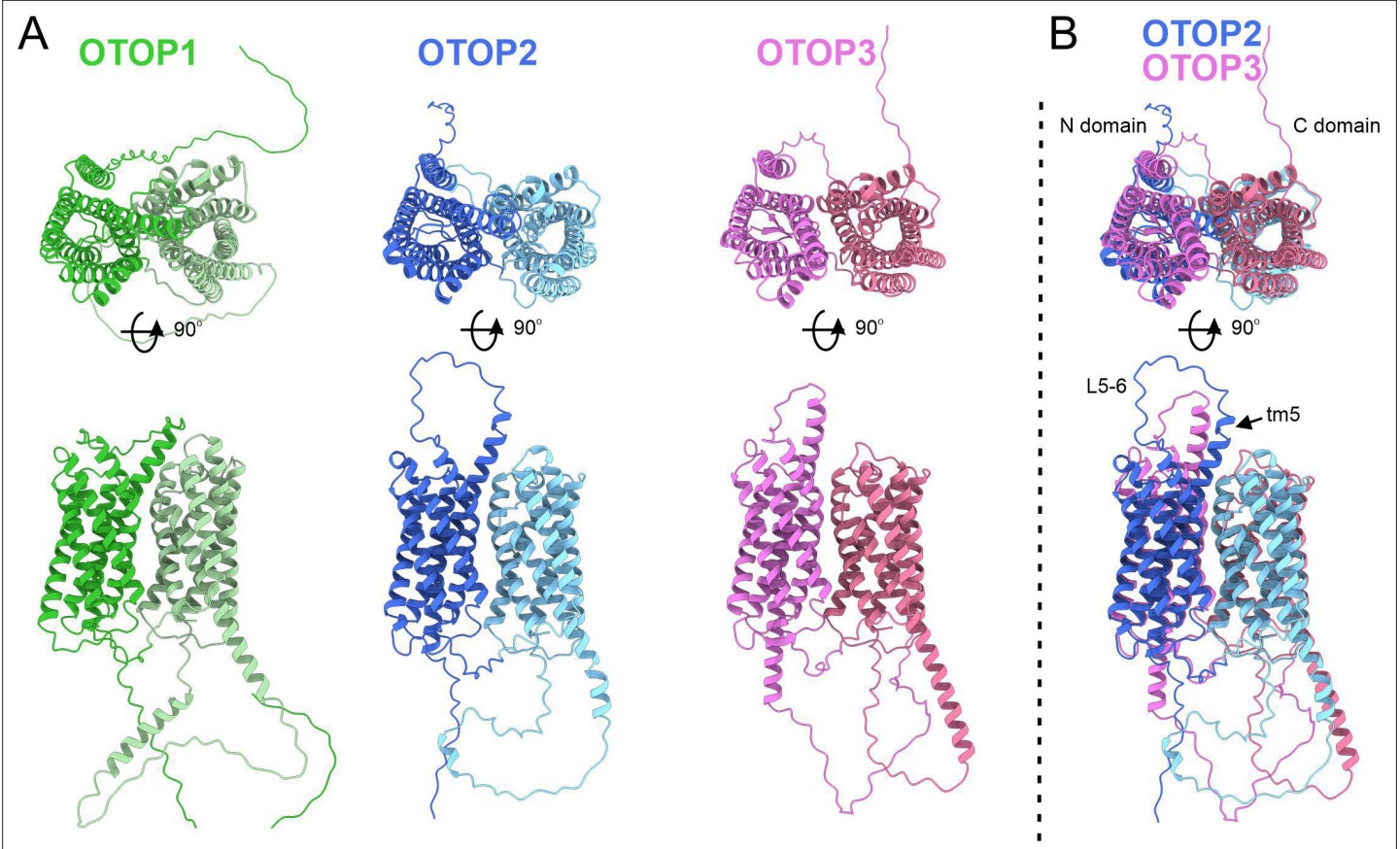

**Figure 8.** Predicted structures of mOTOP1. mOTOP2 and mOTOP3. (**A**) Top views (top) and side views (bottom) of AlphaFold predicted structural models of mOTOP1, mOTOP2, and mOTOP3. The N- and C- domain halves of mOTOP1, mOTOP2, and mOTOP3 are colored green and light green, blue and light blue, and magenta and hot pink, respectively. (**B**) A superimposed overlay of mOTOP2 and mOTOP3 highlights the different orientations of the transmembrane 5 helices and S5-6 linkers.

The online version of this article includes the following figure supplement(s) for figure 8:

**Figure supplement 1.** Comparison between AlphaFold predictions and cryoEM structures.

to gain structural insight into the possible mechanisms by which extracellular linker regions could contribute to gating, we examined the structures of murine OTOP1, OTOP2, and OTOP3 channels predicted by AlphaFold (*Jumper et al., 2021*; *Varadi et al., 2022*). We first verified that the Alpha-Fold structures were reliable by comparing them to the cryoEM structures where available (e.g. for DrOTOP1 and GgOTOP3). This analysis revealed excellent agreement between the observed and expected structures (*Figure 8—figure supplement 1*). Inspection of the predicted structures of the three mammalian OTOP channels shows the putative location of the extracellular linker and associated transmembrane domains (*Figure 8*). Interestingly, the 5–6 linker of OTOP2 is longer, and transmembrane domain 5 (TM5) of OTOP2 adopts a helical structure with a pronounced kink, whereas that of OTOP3 is straight and rod-like. Thus, it is possible that the introduction of the OTOP3 S5-6 linker onto OTOP2, which disrupts conduction at alkaline pH, may do so through a structural rearrangement of transmembrane helices within the N domain of the channel. In contrast, differences between OTOP2 and OTOP3 in the 3–4 and C terminal linkers and associated transmembrane domains are less pronounced, suggesting that single amino acids that differ between the channels may tune the pH dependence of gating. Together, we conclude that extracellular linkers participate in the gating of OTOP channels and that the gating apparatus is distributed across multiple extracellular regions within both the N and C domains.

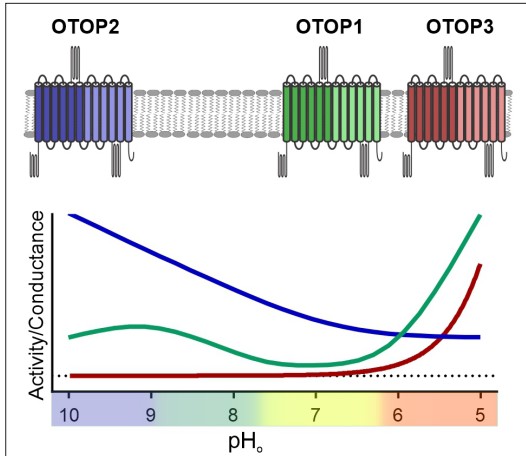

**Figure 9.** pH tuning of OTOP channels. Lowering extracellular pH increases the slope conductance of OTOP1 and OTOP3 channels while it lowers the slope conductance of OTOP2 channels. OTOP1 also has a peak of activity at mildly alkaline pH.

## Discussion

Unlike other ion channels where functional descriptions of functional properties largely preceded their molecular identification, for OTOP1 and related proteins, such descriptions were limited to a few studies of native OTOP1 channels in taste receptor cells (*Chang et al., 2010*; *Bushman et al., 2015*). Thus, while ion selectivity of vertebrate and invertebrate OTOP channels has been well established (*Tu et al., 2018*; *Chen et al., 2019*), descriptions of basic gating mechanisms have been limited. For example, previous studies showed that the channels are not voltage-sensitive (*Tu et al., 2018*), but whether they are ligand-gated was not known. Here we provide the first direct evidence that the three murine OTOP channels are gated by extracellular protons in a subtype-specific manner (*Figure 9*). OTOP1 and OTOP3 show no or little activity at neutral $pH_o$ and are activated by lowering pH, while OTOP2 is active at neutral and alkaline $pH_o$, and lowering $pH_o$ appears to inhibit the channel. The varying pH sensitivity of the three vertebrate OTOP channels may allow them to subserve different functions in the cells within which they are expressed and is reminiscent of the varying temperatures sensitivities of thermo-TRP channels (*Dhaka et al., 2006*).

### Gating in response to changes in extracellular pH

The gating of ion channels by protons has been described previously. For example, the thermosensitive ion channel TRPV1, the acid-sensing ion channels (ASIC), and proton-activated chloride channels (PAC) are all activated by protons acting on extracellular sites (*Waldmann et al., 1997*; *Jordt et al., 2000*; *Lambert and Oberwinkler, 2005*; *Ullrich et al., 2019*; *Yang et al., 2019b*; *Ruan et al., 2020*; *Deng et al., 2021*). Others are gated by protons acting on the intracellular side of the membrane (*Cuello et al., 2010*; *Wang et al., 2010*). Notably, the only other plasmalemma proton channel in eukaryotes, the voltage-gated proton channel Hv1, shows a shift in voltage dependence as a function of pH (*Ramsey et al., 2006*; *Decoursey, 2012*), while a lysosomal $K^+$ channel, TMEM175 was recently shown to function as a proton-activated proton channel (*Cang et al., 2015*; *Hu et al., 2022*; *Zheng et al., 2022*). For channels in which the proton is not the primary permeant ion, the effect of pH on gating can be measured simply by comparing current magnitudes as extracellular pH is varied. In contrast, because changing pH will also change the driving force for proton entry and, therefore, the magnitude of proton currents, establishing an effect of pH (protons) on the gating of a proton channel is more difficult. Such an analysis is further complicated by the exceedingly small conductance of proton channels at physiological or even acid pH (*Decoursey, 2012*), which makes it hard, if not impossible, to measure the effects of pH on the gating of single channels.

The magnitude of OTOP currents was previously shown to vary as a function of $pH_o$ (*Tu et al., 2018*). We confirmed the effects of $pH_o$ and extended the pH range of test solutions, allowing us to observe that OTOP2 and, to a lesser extent, OTOP1 channels can carry outward proton currents in response to extracellular alkalization. The differences in currents suggest underlying differences in gating. To provide direct evidence for differential gating of the vertebrate OTOP channels, we devised a strategy that avoids confounds introduced by the changing driving force for proton movement inherent in these experiments. Taking advantage of differences in the three murine OTOP channels, we provide three independent pieces of evidence that extracellular protons gate the OTOP channels. First, we found that OTOP1 and OTOP3 currents activated slowly in response to acid stimuli, with rates that increased as the extracellular pH was lowered. These results can be explained if lowering the pH opens the channels. Second, the slope conductance of the OTOP channels, which is independent of driving force, varied as a function of $pH_o$ for all three channels.

Consistent with the kinetic data, the slope conductance of both OTOP1 and OTOP3 currents increased dramatically upon lowering the pH below 6.0 (OTOP1) and 5.5 (OTOP3), while the slope conductance of OTOP2 increased with alkalization above pH 8.0. Finally, these results are consistent with pH imaging experiments where proton efflux in response to extracellular alkalinization was only observed for OTOP1 and OTOP2 and not for OTOP3-expressing cells. Together these data allow us to conclude that the three murine OTOP channels are gated differentially by extracellular protons.

## Structure of OTOP channels and role of extracellular linkers

The recent cryo-EM structures of OTOP1 and OTOP3 channels have revealed that the channels adopt a novel fold, with twelve transmembrane alpha-helices organized into two structurally homologous six-helix domains (N and C domains) (*Chen et al., 2019*; *Saotome et al., 2019*). Thus, as a dimer, the channels adopt a pseudo-tetrameric structure. However, in contrast to most ion channels where the subunits come together to form a shared pore, the central cavity of OTOP channels is filled with cholesterol-like molecules and, therefore, cannot mediate ion conduction (*Saotome et al., 2019*). Three putative pathways for protons have been identified in the structures: one each in the N and C domains and another at the interface of N and C domains (intrasubunit interface). Mutations in each putative permeation pathway that cause loss of function have been identified, including mutations of conserved residues (E267 and H574 in ZfOTOP1) at the intrasubunit interface that are required for function but not subunit assembly or trafficking (*Saotome et al., 2019*). Whether these residues mediate gating or permeation of the channels is still not known, and it is possible they participate in both (*Tombola et al., 2008*; *Musset et al., 2011*). Moreover, with three possible routes for proton permeation, there may be multiple pores that open under different conditions. For example, our observation that OTOP1 channels show a biphasic response to extracellular pH could be explained by two pores that open over different pH ranges, a hypothesis that would need to be tested.

Our data also provide a framework for understanding the structural basis for the gating of OTOP channels by protons. Swapping extracellular loops of OTOP2 and OTOP3 selectively changed the gating of the channels in a predictable manner. Swapping loops in the N domain of the channels selectively affected the outward currents elicited in response to alkalinization, such that OTOP2/OTOP3 (L5-6) channels carried less outward current, while OTOP3/OTOP2(L3-4) channels carried more outward current than the respective wildtype channels. They also showed rapid activation kinetics to acid stimuli like OTOP2. This suggests that the N domain either contains the pore that is active over a broad pH range or contains a sensor that opens a common pore over a large pH range (for example, at the intrasubunit interface). In contrast, swapping loops within the C domain had no discernable effect on OTOP2 but changed the gating of OTOP3 channels such that they opened faster and at higher pH (like OTOP2). This suggests that activation of OTOP3 may involve a pH-dependent conformational change within the C domain linkers that otherwise inhibit its activity.

How the structure of the extracellular linkers participate in gating and whether they function as pH sensors or allosterically modulate gating remains to be determined. Structural predictions of MmOTOP2 from AlphaFold (*Jumper et al., 2021*; *Varadi et al., 2022*) point to intriguing differences in the position of S5 domain and the S5-6 linker between OTOP2 and OTOP3 that may underlie differences in gating. In contrast, AlphaFold predicted similar conformations of the L3-L4 loop for both channels, so that we do not presently have a structural basis for the phenotype of the OTOP3/OTOP2(L3-4) chimera. It is worth noting that the chemical makeup of this linker is very variable; for example, OTOP3 has three positively charged residues (two His and one Lys) and three negatively charged residues, while OTOP2 has just one positively and one negatively charged residues (conserved in OTOP1) and several aromatic residues.

Together our description of the differential gating of the three mammalian OTOP channels by protons provides the basis for a deeper understanding of this new ion channel family and will lead to a better understanding of how OTOP channels contribute to a wide range of physiological processes, from gravity senses and taste to digestion, both in health and disease. Further understanding of the mechanism by which protons gate OTOP channels will require a combination of structure-guided mutagenesis and the resolution of the structure of OTOP channels in different states.

# Materials and methods

**Key resources table**

| Reagent type (species) or resource | Designation | Source or reference | Identifiers | Additional information |
|---|---|---|---|---|
| Gene (*M. musculus*) | *Otop1*, *Otop2* and *Otop3* | *Tu et al., 2018*. PMID:29371428 | | |
| Cell line (*Homo-sapiens*) | HEK293 | ATCC | CRL-1573 | |
| Cell line (*Homo-sapiens*) | PAC-KO HEK293 cells | *Yang et al., 2019a*. PMID:31023925 | | |
| Recombinant DNA reagent | *Otop1*, *Otop2*, and *Otop3* in pcDNA3.1 | *Tu et al., 2018*. PMID:29371428 | | |
| Recombinant DNA reagent | *Otop1*, *Otop2*, and *Otop3* – GFP | *Saotome et al., 2019*. PMID:31160780 | | |
| Recombinant DNA reagent | mO2_O3 loop swap mutations | This paper | | cDNAs encode chimeric channels (see methods and *Figure 6—figure supplement 1*). Available upon request |
| Recombinant DNA reagent | mO3_O2 loop swap mutations | This paper | | cDNAs encode chimeric channels (see methods and *Figure 6—figure supplement 1*). Available upon request |
| Recombinant DNA reagent | pHluorin in pcDNA3 | *Miesenböck et al., 1998*. PMID:9671304 | | |
| Chemical compound, drug | CHES | Sigma | C2885 | |
| Chemical compound, drug | PIPES | Sigma | P6757 | |
| Chemical compound, drug | Homopiperazine-1,4-bis(2-ethanesulfonic acid) | Sigma | 53,588 | |
| Software, algorithm | GraphPad Prism 8 and 9 | GraphPad | RRID:SCR_002798 | |
| Software, algorithm | pClamp and clampfit | Molecular Devices | RRID:SCR_011323 | |
| Software, algorithm | Origin | OriginLab corporation | RRID:SCR_002815 | |
| Software, algorithm | CorelDraw | Corel | RRID:SCR_014235 | |
| Software, algorithm | SimplePCI | HCImage | https://hcimage.com/simple-pci-legacy/ | |

## Clones, cell lines, and transfection

Mouse Otop1, Otop2, and Otop3 cDNAs were cloned into the pcDNA3.1 vector as previously described (*Tu et al., 2018*). For experiments in *Figures 6 and 7*, both mOtop2 and mOtop3 cDNAs were cloned into the pcDNA3.1 vector with an N-terminal fusion tag consisting of an octahistidine tag followed by eGFP, a Gly-Thr-Gly-Thr linker, and 3 C protease cleavage site (LEVLFQGP) as previously described (*Saotome et al., 2019*). Chimeras were generated using In-Fusion Cloning (Takara Bio) and were confirmed by Sanger sequencing (Genewiz).

HEK293 cells were purchased from ATCC (CRL-1573). PAC-KO cells were a kind gift from Dr. Zhaozhu Qiu (*Yang et al., 2019b*). The cells were cultured in a humidified incubator at 37 C in 5% $CO_2$ and 95% $O_2$ using a high glucose DMEM (ThermoFisher) containing 10% fetal bovine serum (Life Technology) and 1% Penicillin-streptomycin antibiotics. Cells were passaged every 3–4 days. Cells were tested and found free of mycoplasma using a PCR detection kit (Sigma-Aldrich, USA).

Cells used for patch-clamp recordings were transfected in 35 mm Petri dishes with ~600 ng DNA and 2 µL TransIT-LT1 transfection reagents (Mirus Bio Corporation) following the manufacturer's protocol. OTOP channels were co-transfected with GFP or pHluorin at a ratio of 5:1. The cells were

lifted using Trypsin-EDTA 24 h after transfection, plated onto a coverslip, and used within 3–4 hr for patch-clamp recordings.

## Patch-clamp electrophysiology

Whole-cell patch-clamp recording was performed as previously described (*Tu et al., 2018*). Currents were filtered at 1 KHz and digitized at 5 kHz using an Axonpatch 200B amplifier and Digidata 1322 a 16-bit data acquisition system. Acquisition control and analysis were performed with pClamp 8.2 and Clampfit 9 (Molecular Devices). Patch pipettes with a resistance of 2–4 MΩ were fabricated from borosilicate glass (Sutter instrument) and fire polished. Only cells with stable giga-ohm seals were used for data collection and subsequent analysis. Records were analog filtered at 1 kHz before digital sampling at 5 kHz. Millisecond solution exchange was achieved with a fast-step perfusion system (Warner instrument, SF-77B) custom modified to hold seven microcapillary tubes in a linear array.

The holding potential was –80 mV unless otherwise indicated. For experiments in *Figure 5*, the voltage was held at –80 mV and ramped to +80 m at 1 V/s once per second. The slope of the first ramp after the peak current in response to acids was measured to determine the conductance.

## Patch-clamp electrophysiology solutions

Tyrode's solution contained 145 mM NaCl, 5 mM KCl, 1 mM MgCl2, 2 mM CaCl2, 20 mM dextrose, 10 mM HEPES (pH adjusted to 7.4 with NaOH). Standard pipette solution contained 120 mM Cs-aspartate, 15 mM CsCl, 2 mM Mg-ATP, 5 mM EGTA, 2.4 mM CaCl2 (100 nM free Ca2+), and 10 mM HEPES (pH adjusted to 7.3 with CsOH; 290 mosm). Standard Na +free extracellular solutions contained 160 mM NMDG-Cl, 2 mM CaCl2, and 10 mM buffer based on pH (CHES for pH 10–9, HEPES for pH 8–7.4, PIPES for pH 7, Bis-tris for pH 6.5, MES for pH 6–5.5 and HomoPIPES for pH 5–4.5).

For sodium solution in *Figure 2A*, 160 mM NMDG-Cl was replaced by an equimolar concentration of NaCl. For the low chloride solution in *Figure 2B*, 120 mM HCl was replaced by methane sulfonic acid (CH3SO3H). In these experiments, 200 µM Amiloride was added to block endogenous ASIC channels.

## pH imaging for transfected HEK-293 cells

HEK-293 cells were cultured in 35 mm Petri dishes. OTOP channels and the pH-sensitive indicator pHluorin were co-transfected into the cells. After 24 hr, the cells were lifted and plated on poly-D-lysine-coated coverslips at room temperature. The cells were incubated in standard Tyrodes' solutions before experiments, and the pHluorin fluorescence intensity in response to different solutions was measured. All stimulating solutions were modified from the standard Tyrodes' solution, containing 10 mM of different buffer salt based on the pH (CHES for pH 8.5, MES for pH 6.0, Homo-PIPES for pH 5.0, and Acetic acid for pH 5.0 [HOAC] group). Excitation was 488 nm, and emission was detected at 510 nm using a U-MNIBA2 GFP filter cube (Olympus). Images were acquired on a Hamamatsu digital CCD camera attached to an Olympus IX71 microscope using Simple PCI software. The fluorescence intensity of each cell was normalized to its baseline in Tyrodes' solutions (F0) before the first test stimulus was given to the cells.

## Quantification, statistical analysis, and generation of AlphaFold models

All data are presented as mean ± SEM if not otherwise noted. Statistical analysis was performed using Graphpad Prism 8 or 9 (Graphpad Software Inc). The sample sizes of 3–10 independent recordings from individual cells per data point are similar to those in the literature for similar studies. All data are biological replicates. In some cases where technical replicates were performed (e.g. re-running the test protocol a second time on the same cell), we typically used the first series unless there was a loss of seal resistance and recovery, in which case we used the second replicate. Selection, in this case, was blind to the result/outcome. Representative electrophysiology traces shown in the figures were acquired with pClamp, and in some cases, the data was decimated 10-fold before exporting into graphic programs, Origin (Microcal) and Coreldraw (Corel).

The predicted structural models of mOTOP1, mOTOP2, and mOTOP3 were downloaded from the AlphaFold Protein Structure Database https://alphafold.ebi.ac.uk (*Jumper et al., 2021*; *Varadi et al., 2022*). Figures of the models were generated using UCSF ChimeraX https://www.rbvi.ucsf.edu/chimerax/ (*Goddard et al., 2018*).

## Acknowledgements

We thank Jackson Walker and Anne Tran for expert technical support and all members of the Liman and Ward labs for helpful discussions. We also thank M Goldschen-ohm for critical reading of the manuscript and Z Qui for generously sharing cell lines.

## Additional information

### Funding

| Funder | Grant reference number | Author |
|---|---|---|
| National Institute of General Medical Sciences | R01GM131234 | Emily R Liman |
| National Institute on Deafness and Other Communication Disorders | R01DC013741 | Emily R Liman |

The funders had no role in study design, data collection and interpretation, or the decision to submit the work for publication.

### Author contributions

Bochuan Teng, Conceptualization, Formal analysis, Investigation, Visualization, Methodology, Writing – original draft; Joshua P Kaplan, Conceptualization, Formal analysis, Investigation, Writing – review and editing; Ziyu Liang, Zachary Krieger, Investigation; Yu-Hsiang Tu, Conceptualization, Investigation, Visualization; Batuujin Burendei, Investigation, Visualization, Writing – review and editing; Andrew B Ward, Supervision, Visualization, Writing – review and editing; Emily R Liman, Conceptualization, Formal analysis, Supervision, Funding acquisition, Visualization, Methodology, Writing – original draft, Project administration, Writing – review and editing

### Author ORCIDs

Andrew B Ward http://orcid.org/0000-0001-7153-3769
Emily R Liman http://orcid.org/0000-0003-4765-5496

### Decision letter and Author response

Decision letter https://doi.org/10.7554/eLife.77946.sa1
Author response https://doi.org/10.7554/eLife.77946.sa2

## Additional files

### Supplementary files

• Transparent reporting form

### Data availability

Figure 1-7 - Source Data -1-7 contain the numerical data used to generate the figures.

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
