## [Editor Report]

The manuscript shows that OTOP proton channels are proton-gated with distinct pH sensitivities, and identifies regions on the proteins that alter pH-dependent gating. The main claims are well supported by the data. These findings are likely to be of interest to researchers studying acid/base physiology, sensory physiology, and ion channel biophysics.

---

## [Decision Letter]

**Decision letter after peer review:**

Thank you for submitting your article "Structural motifs for subtype-specific pH-sensitive gating of vertebrate otopetrin proton channels" for consideration by *eLife*. Your article has been reviewed by 3 peer reviewers, including Jon Sack as the Reviewing Editor and Reviewer #1, and the evaluation has been overseen by Richard Aldrich as the Senior Editor. The following individual involved in the review of your submission has agreed to reveal their identity: Ian Scott Ramsey (Reviewer #3).

Essential revisions:

1) Provide additional data to assess pH at the cell membrane during whole-cell patch-clamp experiments. It seems important to distinguish whether channel responses, such as the current decay are caused by pH changes or channel gating. See comments by reviewers #1 and #3. We suggest this could be achieved by: measurement of OTOP current reversal potentials at a series of time points during the rise and fall of OTOP currents; replacement of aspartate in the pipette with an anion that is less likely to be subject to transmembrane diffusion in its neutral form; changing the intracellular pH.

2) Provide additional data to establish what solution exchange times are. See comments by reviewers #1 and #3. We suggest this could be achieved by open tip potential measurements.

3) The section describing the effect of extracellular pH on slope conductance needs to be revised and the value of the slope conductance measurements clarified. See comments by reviewers #2 and #3.

4) Discuss which titratable residues could potentially be pH sensors. Experiments with point mutations of titratable residues could greatly increase the impact of this study.

5) Address each of the comments of the reviewers by revision of the manuscript or explain in the response letter why not.

*Reviewer #1 (Recommendations for the authors):*

Abstract

line 5 "permeated by"?

Results

line 75 "the rate of current decay was faster as the pHo was lowered." Could this be a result of H^+^ accumulation/depletion?

line 83 "this suggests that channels had opened in response to the pH 9 stimulus and subsequently closed over a period of several seconds upon return to neutral pH. In response to pH 8 and 7 solutions, no change in the baseline currents was observed." Could this be a result of H^+^ accumulation/depletion?

line 144 "Notably, the response of OTOP3-expressing cells was significantly slower than that of OTOP1 or OTOP2-expressing cells." I suggest more clearly pointing out in Figure 3 the basis for this conclusion.

Figure 4D – A log Y scale might help readers assess the relative pH-dependence of tau-off.

line 164 "Note that we excluded the first few milliseconds where responses deviated from an exponential time course, likely due to the non-instantaneous rate of solution exchange." Confusing, as the manuscript seems to soon state that the 15ms tau was the time of solution exchange.

line 183 "the deactivation rate of the currents upon return to neutral pH did not vary as a function of pH for any of the three channels and instead reflected the exchange time for the solutions, as expected (Figure 4D). " Independent evidence of the exchange time is needed.

Figure 5 – The repetitive ramp currents in Figure 5A as well as the reversal potentials in slope-conductance measurements seem consistent with H^+^ accumulation in patched cells. Assessment of whether H^+^ accumulation/depletion impacts currents seems needed. Could H^+^ accumulation in cells account for all of the current decay after acid stimuli?

Figure 5C – n-values missing.

Figure 9 – What does 1.0 conductance mean? Why doesn't OTOP3 go to 1.0? Could it be that all OTOPs have lower single-channel conductance at low pH, but this is obscured by the increase in Popen at lower pH for OTOP1 and 3?

*Reviewer #2 (Recommendations for the authors):*

This is an interesting manuscript investigating the basic mechanism of OTOPs gating. It provides evidence that protons gate the channels in a subtype-specific manner and identifies two extracellular loops in the protein as important contributors to pH-dependent gating. Overall, the experiments are well justified, and the results are explained clearly. The conclusions are supported by the data but there are several points that need to be addressed.

The section describing the effect of extracellular pH on slope conductance needs to be revised. Slope conductance is defined as dI/dV and can be positive or negative depending on the voltage range of activation of the channel and the current reversal potential. The connection between slope conductance and N*Po*g indicated by the authors is unclear, as Po is usually the open probability at a given V, and g is the single channels chord conductance. If the authors really wanted to measure a parameter that does not depend on the driving force for proton entry, they should have just calculated the conductance as G = I/(V-Erev), not I/V.

The I/V ratio for OTOP2 varies with pHo in a confusing way (Figure 5C). It decreases going from pHo = 10 to pHo = 6, and then increases going to pHo = 5 (while the current remains constant between pHo = 6 and pHo = 5, Figure 1C). This seems inconsistent with the statement that "OTOP2 is constitutively open" which implies that Po does not depend on pH in this channel.

The authors state that the pH imaging data in Figure 3 provide the first direct evidence that the three murine OTOP channels are gated by extracellular protons in a subtype-specific manner. But what about the current data in Figure 1. Why is that data not sufficient to make the claim?

In Figure 4A, for OTOP1 and OTOP3, a k(on) connects a closed state in low H^+^ conc. to an open state in high H^+^ conc., but the k(off) does not connect the open state to the closed state. So, in what state is the channel right after returning to neutral pH? Did the authors try to open the channels with two consecutive acid pulses?

The structural differences in the L5-6 loops of the three OTOPs predicted by AlphaFold are interesting but what about the L3-4 loops? How are these differences helpful in understanding the experimental data on loop swapping? How do AlphaFold predictions compare with the Cryo-EM structures in the resolved transmembrane region?

While the 3D structure of the loops is expected to affect function, what about the number/nature of charged and titratable residues? For instance, the L3-4 loop of OTOP2 contains three positive charges, of which two are His residues, while OTOP1 and OTOP2 have a net negative charge carried by a conserved Glu residue. It would be useful to discuss this kind of difference in light of the results of loop swapping.

*Reviewer #3 (Recommendations for the authors):*

1. Currents shown in Figure 1 at nearly all pHo tested show a pronounced decay that is reported to be faster when pHo is lower (lines 74-76). However, it is not clear whether currents plotted in Figure 1C were measured at the peak vs. some other time during the pHo change. Do the authors do not offer any insight into the mechanism underlying current decay? For example, the relatively large current amplitudes suggest that despite pipette solution pH buffering, pHi could change as a result of current flowing through open Otop channels; alternatively, Otops might exhibit a form of desensitization or inactivation. If the former is the case, are the authors confident that current amplitudes are accurately measured and that the imposed change in pHo has been achieved at the time when currents were measured? (See also comments about solution exchange time below).

2. Intracellular pH is 7.4 in all experiments (except Figure 2, where only Otop2 was measured), and the effect of changing pHi does not appear to have been tested. Why not?

3. pHluorin imaging shown in Figure 3 is indirect and, due to lack of voltage clamp and lack of good pHi buffering, an intrinsically less robust method than electrophysiology; perhaps this data is better suited to Supplementary material? Experiments demonstrating net inward vs. outward proton fluxes during voltage ramps applied at various pHo (Figure 5B) appear to be a more appropriate method for tracking the direction of permeant ion movement.

4. The fastest gating transition measured in Figure 4 is limited by solution exchange time, as the authors note, but it's not clear that solution exchange times were independently measured (i.e., by open-tip potential changes) using the same apparatus. Is the off-time constant for Otop2 (32 ms; Figure 4C, middle panel) substantially slower than the solution exchange time (if so, then the simplified scheme proposed in Figure 4A would not hold for Otop 2)?. Alternatively, the relatively slow current deactivation rate measured in Figure 4C might not be representative of the average behavior (as suggested by summary data shown in Figure 4D, middle panel). Furthermore, the kinetic schemes shown in Figure 4A do not consider that proton binding or unbinding from its putative binding site(s) might be rate-limiting for gating transitions; is this assumption supported by the experimental data?

5. What new information is gleaned from measuring slope conductance (Figure 5C)? The G vs. pHo graphs appear to have the same shape as I vs. pHo graphs in Figure 1C, suggesting that the two methods report the same pH-dependent gating transitions. Could channel numbers be estimated (i.e., via current variance analysis) to calculate unitary conductances for Otop 1-3?

6. Otop2/3 chimera data (Figure 6) are interesting, but leave open the question as to whether discrete titratable residues are responsible for low- vs. high-pH dependent gating. It would seem that the chimeras would help narrow the range of candidates, allowing the authors to focus on just a few. Additional data in single- or double-mutant channels would greatly enhance the strength of the manuscript. Why are no Otop1 chimeras shown?

7. The reliability of AlphaFold-generated models is not independently established (i.e., during MD simulations), and it remains unclear whether the structural differences shown in Figure 8 are truly representative of physiologically meaningful differences in protein structure. How does the data in Figure 8 contribute to knowledge about the structural basis of pH-dependent gating in Otop channels? Do models identify specific titratable residues that could be experimentally tested (see item 6 above)?

---

## [Author Response]

Essential revisions:1) Provide additional data to assess pH at the cell membrane during whole-cell patch-clamp experiments. It seems important to distinguish whether channel responses, such as the current decay are caused by pH changes or channel gating. See comments by reviewers #1 and #3. We suggest this could be achieved by: measurement of OTOP current reversal potentials at a series of time points during the rise and fall of OTOP currents; replacement of aspartate in the pipette with an anion that is less likely to be subject to transmembrane diffusion in its neutral form; changing the intracellular pH.

We now provide additional data to assess pH at the cell membrane and its contribution to current decay (Figure 5- supplementary figure 1).

Figure 5- supplementary figure 1A, B shows that Erev shifts negative for OTOP1 and OTOP2 during the pH 5 acid stimuli (consistent with acid loading of the cells). For OTOP3 currents, which are slowly activating (over several seconds), the shift in Erev is in the positive direction, consistent with a larger fraction of the current coming from OTOP channels (as compared with leak current) at later time points. From these experiments, it is clear that the intracellular pH decreases during the acid stimuli for OTOP1 and OTOP2. It is for this reason that we collected data for OTOP1 and OTOP2 at early time points (usually at 1 second after the onset of the stimulus for I-V curves).

To determine if the change in intracellular pH might account for the decay of the currents in response to sustained stimuli, we compared the observed magnitudes of the currents with that expected if the driving force for proton entry changed as measured from Erev. We used the equation: I(t) = I(1) * (Vm-Erev(t))/(Vm-Erev(1)), where I(1) and I(x) are the currents at times 1 second (the first measurement we could make) and x seconds, Erev (1) and Erev(x) are the reversal potential of the current at times 1 and x, and Vm was the membrane potential at which I was measured (-80 mV for these experiments). The results are shown in Figure 5- supplementary figure 1C. We can account for most of the decay of the OTOP1 currents by the change in driving force (this is also evident in panel A, where the slope of the currents changes little during the stimulus). In contrast, we noted that there was a large deviation for OTOP2 currents between the magnitudes of the currents as predicted from the shift in Erev, and what was measured. This suggests that the decay of OTOP2 currents might be attributed to changes in gating during the stimulus application. Whether this reflects an inactivation of the channels or inhibition by intracellular pH remains an open question.

Note that we do not agree that aspartate as the anion in the intracellular solution is a concern as its pKa is 3.9 and previously we did not observe the intracellular pH to drop below pH 5 in unperturbed cells (Tu et al., 2018). Moreover, if aspartate did shuttle protons out of the cell, this would help stabilize the intracellular pH in response to the acid stimuli. We have also tested the effects of changing pHi on whole-cell OTOP currents, but as these experiments require additional validation (e.g. with excised patches) and are not central to the main argument of the manuscript, that the channels are gated by extracellular protons acting on linker domains, we feel adding this data would detract from the manuscript.

2) Provide additional data to establish what solution exchange times are. See comments by reviewers #1 and #3. We suggest this could be achieved by open tip potential measurements.

This is a good point and we now included data that allowed us to estimate the kinetics of the solution exchange (Figure 4B,C). Note that the open tip method works for excised patches but is not a good estimate of solution exchange time for whole-cell recording, where the cell will disrupt laminar solution flow and slow down exchange rates (Li et al., 2000). We, therefore, adopted a variation of the method described in Li et al., 2000, estimating solution exchange time using the response of an open K^+^ channel (KIR2.1) to changes K^+^ concentrations. In these experiments, the K^+^ channel was either transfected alone or co-transfected with OTOP2, to allow measurement of currents through both channels in the same cell (in response to different solutions). The results show that OTOP2 currents are activated as quickly as the K^+^ currents, consistent with an open channel, and that decay rates for all channels are not different the expected decay rate due to removal of the permeant ion.

3) The section describing the effect of extracellular pH on slope conductance needs to be revised and the value of the slope conductance measurements clarified. See comments by reviewers #2 and #3.

Reviewer 2 preferred a measurement of slope conductance as I/(V-Vrev). We now explain that we measured DI/DV = ((I1-I2)/(V1-V2)) which is formally identical to I/(V-Vrev) for the case where V2=Vrev (so I2=0). Reviewer #3 asked “What new information is gleaned from measuring slope conductance” as compared to measuring current (at V = -80 mV).” We now explain that measuring slope conductance allows us to assess channel activity when the holding potential is close to Erev, and it normalizes for driving force.

Lines 203-211 now read “The slope conductance (DI/DV) was measured from response to ramp depolarizations at or near the peak of the response to the stimulus (pH_o_ = 10 to 5) (Figure 5A, B) using a range of V_m_ from -80 mV to 0 mV, where the currents were generally linear and not contaminated by endogenous currents. Slope conductance (G) is related to the open probability (Po) of the channel by G = N*g*Po, where N is the number of channels and g is the single-channel conductance. Since the number of channels is constant, the slope conductance is a measure of Po*g. Importantly, this provides a measure of open probability, even where the holding potential is close to the equilibrium potential for H^+^ and therefore currents are small (e.g. for stimuli at 8.0).

4) Discuss which titratable residues could potentially be pH sensors. Experiments with point mutations of titratable residues could greatly increase the impact of this study.

A main conclusion of the paper is that the gating apparatus is distributed across multiple (if not all) extracellular linkers, which contain dozens of titratable residues. Thus, until we are able to resolve more clearly (e.g. with cryo-EM) the gating transition, we believe that discussion of such residues is premature. Moreover, we have found in unpublished work that mutations of residues that are not considered titratable can change pH-dependent gating, underscoring our hesitation to make any such claims and highlighting the difficulty in distinguishing between effects on ligand binding (where H^+^ is the ligand) from allosteric effects on gating (e.g. lowing the energy for the channel to open). A limited discussion of some of these points is included in lines 377-386.

“How the structure of the extracellular linkers participate in gating and whether they function as pH sensors or allosterically modulate gating remains to be determined. Structural predictions of mOTOP2 from AlphaFold (Jumper et al., 2021; Varadi et al., 2022) point to intriguing differences in the position of S5 domain and the S5-6 linker between OTOP2 and OTOP3 that may underlie differences in gating. In contrast, AlphaFold predicted similar conformations of the L3-L4 loop for both channels so that we do not presently have a structural basis for the phenotype of the OTOP3/OTOP2(L3-4) chimera. It is worth noting that the chemical makeup of this linker is very variable; for example, OTOP3 has three positively charged residues (two His and one Lys) and three negatively charged residues, while OTOP2 has just one positively and one negatively charged residue (conserved in OTOP1) and several aromatic residues. “

5) Address each of the comments of the reviewers by revision of the manuscript or explain in the response letter why not.

We have thoroughly revised the manuscript to address the reviewer’s comments.

Reviewer #1 (Recommendations for the authors):Abstractline 5 "permeated by"?

Changed as suggested.

Resultsline 75 "the rate of current decay was faster as the pHo was lowered." Could this be a result of H^+^ accumulation/depletion?

Yes, we now add the following to indicate that this is likely the case, with appropriate references. …” likely due in part to intracellular accumulation of protons (Bushman et al., 2015; De-la-Rosa et al., 2016; DeCoursey and Cherny, 1996; Tu et al., 2018) (see below)”

line 83 "this suggests that channels had opened in response to the pH 9 stimulus and subsequently closed over a period of several seconds upon return to neutral pH. In response to pH 8 and 7 solutions, no change in the baseline currents was observed." Could this be a result of H^+^ accumulation/depletion?

Yes, we have now added “this may reflect cytosolic H^+^ depletion during the pH 9 stimulus, creating a driving force for proton entry through open OTOP1 channels that subsequently close when the pH was restored to 7.4”

line 144 "Notably, the response of OTOP3-expressing cells was significantly slower than that of OTOP1 or OTOP2-expressing cells." I suggest more clearly pointing out in Figure 3 the basis for this conclusion.

“Notably, the response of OTOP3-expressing cells was significantly slower than that of OTOP1 or OTOP2-expressing cells (compare Figure 3C with 3A and 3B).

Figure 4D – A log Y scale might help readers assess the relative pH-dependence of tau-off.

This is a good point. We have now used a split Y-scale to allow the reader to see the pH dependence of tau-off. We preferred this over a log scale which compressed the scale, making it hard to see any trends.

line 164 "Note that we excluded the first few milliseconds where responses deviated from an exponential time course, likely due to the non-instantaneous rate of solution exchange." Confusing, as the manuscript seems to soon state that the 15ms tau was the time of solution exchange.

The solution exchange shows a lag that depends as compared with the signal generated by the computer that depends in part on the distance that needs to be traveled to the perfusion tube. We only eliminated the lag, to fit the current kinetics. We did not eliminate the time for the solution exchange per se. To clarify this point, we changed the text to read “Note that we excluded the first few milliseconds where responses deviated from an exponential time course, due to a lag in the solution exchange.”

line 183 "the deactivation rate of the currents upon return to neutral pH did not vary as a function of pH for any of the three channels and instead reflected the exchange time for the solutions, as expected (Figure 4D). " Independent evidence of the exchange time is needed.

We now show measurements of solution exchange times using an open K^+^ channel and changing the concentration of K^+^ in the solution. See above.

Figure 5 – The repetitive ramp currents in Figure 5A as well as the reversal potentials in slope-conductance measurements seem consistent with H^+^ accumulation in patched cells. Assessment of whether H^+^ accumulation/depletion impacts currents seems needed. Could H^+^ accumulation in cells account for all of the current decay after acid stimuli?

We measured the reversal potentials from experiments as in 5A as a function of time and the results are shown in Figure 5 – supplementary figure 1. As described above, this showed that indeed there is H^+^ accumulation during the 4 s stimulus. For OTOP1, the change in driving force can account for much of the current decay. This is not true for OTOP2. We could not measure this for OTOP3 as in this series of experiments we did not observe a shift in Erev that indicated that H^+^ accumulated and did not observe current decay. In addition to the new supplementary figure, we modified the text to describe these results. See above for more details.

Figure 5C – n-values missing.

Added.

Figure 9 – What does 1.0 conductance mean? Why doesn't OTOP3 go to 1.0? Could it be that all OTOPs have lower single-channel conductance at low pH, but this is obscured by the increase in Popen at lower pH for OTOP1 and 3?

This is a conceptual diagram meant to illustrate the activity/conductance of each channel as a function of pH. We now removed the scale (which was arbitrary). The activity of OTOP3 is lower at the most acidic pH to illustrate the concept that OTOP3 activation is shifted to the right.

Reviewer #2 (Recommendations for the authors):This is an interesting manuscript investigating the basic mechanism of OTOPs gating. It provides evidence that protons gate the channels in a subtype-specific manner and identifies two extracellular loops in the protein as important contributors to pH-dependent gating. Overall, the experiments are well justified, and the results are explained clearly. The conclusions are supported by the data but there are several points that need to be addressed.The section describing the effect of extracellular pH on slope conductance needs to be revised. Slope conductance is defined as dI/dV and can be positive or negative depending on the voltage range of activation of the channel and the current reversal potential. The connection between slope conductance and N*Po*g indicated by the authors is unclear, as Po is usually the open probability at a given V, and g is the single channels chord conductance. If the authors really wanted to measure a parameter that does not depend on the driving force for proton entry, they should have just calculated the conductance as G = I/(V-Erev), not I/V.

The reviewer is correct that for voltage-sensitive ion channels the slope conductance can be negative (e.g. in the range where the channels gate). For OTOP channels, which are not voltage-sensitive (Tue et al., 2018), the slope of the I/V curve (DI/DV) is always positive (or zero) and is not dependent on the reversal potential of the channels, or the voltage. The reason we can use the slope conductance from the macroscopic I-V curve is that Po does not change as a function of voltage (nor does N), and thus it is in principle, a measure of G. Note that we measured (DI/DV) not I/V (this was a typo that has been corrected on line 194), and this is formally the same as I/(V-Erev). Importantly, G as defined by DI/DV does not require that we know Erev (which was out of range for most of the experiments).

The new text reads “The slope conductance (DI/DV) was measured from ramp depolarizations at or near the peak of the response to the stimulus (pHo = 10 to 5) (Figure 5 A, B) using a range of V_m_ from -80 mV to 0 mV, where the currents were generally linear and not contaminated by endogenous currents.”

The I/V ratio for OTOP2 varies with pHo in a confusing way (Figure 5C). It decreases going from pHo = 10 to pHo = 6, and then increases going to pHo = 5 (while the current remains constant between pHo = 6 and pHo = 5, Figure 1C). This seems inconsistent with the statement that "OTOP2 is constitutively open" which implies that Po does not depend on pH in this channel.

We changed the text (Line 207) to read “In OTOP2-expressing cells, the slope conductance was highest in alkaline pH (pH_o_ = 10) and decreased as the pHo was lowered” and in several instances changed the wording to state that OTOP2 is open at neutral pHo. We agree that for OTOP2, the changes in conductance are not straightforward to interpret in terms of gating of the channels by pHo, and that the most conservative interpretation is that the channels are open over a large pH range. This could be considered constitutive activity, which does not imply that it cannot be regulated.

The authors state that the pH imaging data in Figure 3 provide the first direct evidence that the three murine OTOP channels are gated by extracellular protons in a subtype-specific manner. But what about the current data in Figure 1. Why is that data not sufficient to make the claim?

We changed the wording (we presume the reviewer was referring to the discussion). We believe the pH imaging provides important corroboration of a central finding of the paper, that OTOP2, and to a lesser extent OTOP1, mediates proton efflux to alkalinizing pHo. It also describes a methodology that may be useful for future studies of the channels.

In Figure 4A, for OTOP1 and OTOP3, a k(on) connects a closed state in low H^+^ conc. to an open state in high H^+^ conc., but the k(off) does not connect the open state to the closed state. So, in what state is the channel right after returning to neutral pH? Did the authors try to open the channels with two consecutive acid pulses?

We did not study the closing rates of the channels (e.g. with two pulses). Regardless of closing rates, Tau off should reflect the removal of H^+^ as the permeant ion (it is unlikely, although formally possible that closing could be faster than the change in currents due to the reduction in the permeant ion, but we do not have any evidence for this). Regarding the diagram, we considered that this might be more confusing than helpful, and removed it.

The structural differences in the L5-6 loops of the three OTOPs predicted by AlphaFold are interesting but what about the L3-4 loops? How are these differences helpful in understanding the experimental data on loop swapping?

AlphaFold predicted similar conformations in the L3-L4 loop so we do not have a structural basis for the phenotype at this point. This is now described in the manuscript (Lines 291-293)

How do AlphaFold predictions compare with the Cryo-EM structures in the resolved transmembrane region?

This is an excellent question. We now show in Figure 8 – supplementary figure 1 that the TM domains of the AlphaFold predicted and the cryoEM resolved structures show a high degree of similarity, over both the N domain helices and C domain helices. This gives us more confidence in the AlphaFold models and their utility in predicting novel structures (e.g. OTOP2) and predicting regions of the channels that could not be resolved in the cryoEM structures, such as the loops.

While the 3D structure of the loops is expected to affect function, what about the number/nature of charged and titratable residues? For instance, the L3-4 loop of OTOP2 contains three positive charges, of which two are His residues, while OTOP1 and OTOP2 have a net negative charge carried by a conserved Glu residue. It would be useful to discuss this kind of difference in light of the results of loop swapping.

I believe the reviewer meant that OTOP3 (not OTOP2) has two His residues. Of the 12 amino acids that we swapped, 9 varied between OTOP2 and OTOP3. Six of the swaps changed a charged residue. We do not know which of these changes are responsible for the change in gating of OTOP3 and whether the charge on these residues is the relevant parameter. We now mention this in the discussion (Lines 384-386).

Reviewer #3 (Recommendations for the authors):1. Currents shown in Figure 1 at nearly all pHo tested show a pronounced decay that is reported to be faster when pHo is lower (lines 74-76). However, it is not clear whether currents plotted in Figure 1C were measured at the peak vs. some other time during the pHo change. Do the authors do not offer any insight into the mechanism underlying current decay? For example, the relatively large current amplitudes suggest that despite pipette solution pH buffering, pHi could change as a result of current flowing through open Otop channels; alternatively, Otops might exhibit a form of desensitization or inactivation. If the former is the case, are the authors confident that current amplitudes are accurately measured and that the imposed change in pHo has been achieved at the time when currents were measured? (See also comments about solution exchange time below).

The currents were measured at their peak (now stated in the figure legend). Regarding the question of whether the “imposed change in pHo has been achieved at the time when currents were measured?” this is clearly the case for OTOP1 and OTOP3 that have time constants for activation of >100 ms, more than 5-fold slower than the solution exchange times of ~20 ms. For OTOP2, the activation kinetics are similar to the rate of the solution exchange, and they decay rapidly thereafter. Thus, it is possible that the currents might have been larger if the solution exchange was faster (see below). We tried to see if we would get larger OTOP2 currents (meaning we missed the peak) by holding Vm at EH and then changing Vm to -80 mV after the stimulus was applied, but we could only do this for the condition where the difference of pHi and pHo was one pH unit (meaning EH was +60 mV), and this protocol had little effect on the current magnitudes. Thus, we agree that for OTOP2, we might have missed the peak current, but we do not believe that this changes any of the conclusions of the paper.

2. Intracellular pH is 7.4 in all experiments (except Figure 2, where only Otop2 was measured), and the effect of changing pHi does not appear to have been tested. Why not?

We chose to focus on the role of extracellular pH in gating the OTOP channels, and its structural basis as this is an important way in which the channels may be regulated under physiological conditions, and it is more tractable.

The effect of intracellular pH on gating of the channels is an important but thorny issue. As researchers who work on Hv1 well know, controlling the intracellular pH in a cell with large proton currents is notoriously difficult and the problem is compounded for OTOP channels that cannot be closed by voltage to limit proton flux. Moreover, we are not able to do the kind of detailed analyses that we could in measuring effects of extracellular pH (e.g. looking at the kinetics of responses). Thus, while we have done preliminary experiments to measure OTOP currents with solutions of varying intracellular pH, we feel that including these experiments, which are not germane to the main conclusion of the manuscript, would be premature.

3. pHluorin imaging shown in Figure 3 is indirect and, due to lack of voltage clamp and lack of good pHi buffering, an intrinsically less robust method than electrophysiology; perhaps this data is better suited to Supplementary material? Experiments demonstrating net inward vs. outward proton fluxes during voltage ramps applied at various pHo (Figure 5B) appear to be a more appropriate method for tracking the direction of permeant ion movement.

The pHlourin imaging data corroborates data in Figure 1. It also provides a method for observing activity of OTOP channels with fluorometry which may be applicable to investigators working in vivo or with cells that are difficult to patch. As an example of this, note that (Parikh et al., Nature 2019) used imaging approaches to validate expression of OTOP2 in the colon; these investigators can now use alkalizing as well as acidifying stimuli to provide evidence for OTOP2 expression.

4. The fastest gating transition measured in Figure 4 is limited by solution exchange time, as the authors note, but it's not clear that solution exchange times were independently measured (i.e., by open-tip potential changes) using the same apparatus. Is the off-time constant for Otop2 (32 ms; Figure 4C, middle panel) substantially slower than the solution exchange time (if so, then the simplified scheme proposed in Figure 4A would not hold for Otop 2)?. Alternatively, the relatively slow current deactivation rate measured in Figure 4C might not be representative of the average behavior (as suggested by summary data shown in Figure 4D, middle panel). Furthermore, the kinetic schemes shown in Figure 4A do not consider that proton binding or unbinding from its putative binding site(s) might be rate-limiting for gating transitions; is this assumption supported by the experimental data?

We now included data that allows us to estimate the kinetics of the solution exchange in a realistic scenario – that is in whole-cell recording, using an open K^+^ channel (KIR2.1) and changes in potassium concentrations (see above). Please note that there was some variation from cell to cell in the solution exchange time (as estimated with the K^+^ channel), likely due to the placement of the cell in relation to the solution flow through the microcapillary tubes (three of the nine cells had similarly slow decay rates).

5. What new information is gleaned from measuring slope conductance (Figure 5C)? The G vs. pHo graphs appear to have the same shape as I vs. pHo graphs in Figure 1C, suggesting that the two methods report the same pH-dependent gating transitions. Could channel numbers be estimated (i.e., via current variance analysis) to calculate unitary conductances for Otop 1-3?

We now more clearly explain that the slope conductance allows a measurement of gating near the neutral pH, where driving force is diminished and thus currents will not be detected. The fact that the conductance measurements and current measurements are related (but not the same) should be reassuring. Regarding estimating single-channel conductance with noise analysis, this is difficult for a channel that is gated by its permeant ion (usually this experiment is done by varying Po, by changing the voltage, while keeping the permeant ion concentration fixed). Moreover, while it would be interesting to know the single-channel conductance of the OTOPs, this would not change the interpretation of the data or the main conclusion of the paper.

New text reads “Slope conductance is related to the open probability (Po) of the channel by G = N*g*Po, where N is the number of channels and g is the single-channel conductance. Since the number of channels is constant, the slope conductance is a measure of Po*g. Importantly, this provides a measure of open probability, even where the holding potential is close to the equilibrium potential for H^+^ and therefore currents are small (e.g. for stimuli at 8.0).” (lines 206-211).

6. Otop2/3 chimera data (Figure 6) are interesting, but leave open the question as to whether discrete titratable residues are responsible for low- vs. high-pH dependent gating. It would seem that the chimeras would help narrow the range of candidates, allowing the authors to focus on just a few. Additional data in single- or double-mutant channels would greatly enhance the strength of the manuscript. Why are no Otop1 chimeras shown?

Because we did not find one extracellular loop to be a key determinant of gating, and rather found that gating in response to changes in pH are distributed among the loops, our results do not allow us to hone in on single amino acid residues. Indeed, we do not believe such residues exist, but rather the “pH sensor” may consist of multiple titratable residues, not necessarily on the same linker. OTOP1 was not used in the chimera study because its functional properties are intermediate, so that it would be harder to resolve a change in gating in chimeras with OTOP2 or OTOP3.

7. The reliability of AlphaFold-generated models is not independently established (i.e., during MD simulations), and it remains unclear whether the structural differences shown in Figure 8 are truly representative of physiologically meaningful differences in protein structure. How does the data in Figure 8 contribute to knowledge about the structural basis of pH-dependent gating in Otop channels? Do models identify specific titratable residues that could be experimentally tested (see item 6 above)?

We have now tested the reliability of the Alphafold model by comparing them with the cryoEM structures, and they perform very well (Figure 8 – supplementary figure 1). The main take-home message from the comparison of the structures in Figure 8 is that they predict a large difference in the overall topology of the S5-6 linker between mOTOP2 and mOTOP3. We prefer to include these models in the manuscript, and let the reader decide if they are useful.